# ARE LARGE VISION LANGUAGE MODELS GOOD GAME PLAYERS?

**Xinyu Wang[1], Bohan Zhuang[2], Qi Wu[1]**
[1]The University of Adelaide, Australia
[2]Zhejiang University, China
{xinyu.wang02, qi.wu01}@adelaide.edu.au

## ABSTRACT

Large Vision Language Models (LVLMs) have demonstrated remarkable abilities in understanding and reasoning about both visual and textual information. However, existing evaluation methods for LVLMs, primarily based on benchmarks like Visual Question Answering and image captioning, often fail to capture the full scope of LVLMs' capabilities. These benchmarks are limited by issues such as inadequate assessment of detailed visual perception, data contamination, and a lack of focus on multi-turn reasoning. To address these challenges, we propose LVLM-Playground, a game-based evaluation framework designed to provide a comprehensive assessment of LVLMs' cognitive and reasoning skills in structured environments. LVLM-Playground uses a set of games to evaluate LVLMs on four core tasks: Perceiving, Question Answering, Rule Following, and End-to-End Playing, with each target task designed to assess specific abilities, including visual perception, reasoning, decision-making, etc. Based on this framework, we conduct extensive experiments that explore the limitations of current LVLMs, such as handling long structured outputs and perceiving detailed and dense elements. Code and data are publicly available at https://github.com/xinke-wang/LVLM-Playground.

## 1 INTRODUCTION

Large Vision Language Models (LVLMs) (Liu et al., 2024a; Bai et al., 2023; Zhu et al., 2024) have recently demonstrated remarkable capabilities in processing and generating visual and linguistic information. These models extend the strengths of Large Language Models (LLMs) by incorporating the ability to understand and reason about visual data alongside textual inputs, exhibiting human-like reasoning capabilities across a variety of complex tasks that demand strong reasoning skills.

However, the current evaluation methods for LVLMs vary significantly across different studies, lacking a unified criterion. Most existing LVLM works are assessed using multimodal data established before the LVLM era, such as Visual Question Answering (VQA) (Antol et al., 2015), image captioning (Chen et al., 2015), Optical Character Recognition (OCR) (Fu et al., 2024), etc. While these tasks have been instrumental in assessing model performance, they present several limitations:

- **Inadequate Assessment of Detail Perception.** Existing evaluation tasks, such as generic VQA and image captioning, often fail to effectively assess models' ability to perceive and understand fine-grained visual details. For example, in some VQA datasets, questions can be correctly answered without referencing the accompanying images, relying solely on text-based world knowledge (Chen et al., 2024a). This is because some of these questions are not even genuine visual dependent, and thus they might not fully evaluate the models' visual perception abilities.

- **Risk of Data Contamination.** The extensive training of LVLMs on vast amounts of data from diverse sources increases the risk of data contamination, where training data includes some of the question-answer pairs used for testing (Dong et al., 2024; Zhang et al., 2024; Wei et al., 2023). Such overlap can lead to overestimated performance on tasks like VQA, as models may recall answers from their training data rather than genuinely understanding and reasoning about inputs.

- **Limitations of Metrics.** Existing metrics for evaluating LVLMs on generative tasks such as CIDEr and CLIP Score, heavily rely on short, predefined captions. These metrics are not designed to evaluate the rich and varied outputs that LVLMs can produce, especially for open-ended questions without unique correct answers. This limitation forces many LVLM methods to prompt the models to generate short answers to fit these metrics (Li et al., 2023), which can restrict the models' ability to showcase their full capabilities.

- **Inconsistent Prompts.** LVLMs often employ specific prompts during training and inference, including zero-shot, few-shot, and Chain-of-Thought (CoT) prompting. These variations can significantly impact performance (Wei et al., 2022; Li et al., 2024b), making it difficult to attribute differences to the models themselves rather than the prompts used.

- **Neglect of Multi-turn/Long-context Reasoning.** Current evaluation benchmarks focus predominantly on single-turn interactions, overlooking the importance of multi-turn reasoning and long-context understanding in LVLMs (Liu et al., 2024b). While real-world applications often require engaging in extended dialogues and maintaining context over multiple exchanges, these capabilities are not adequately assessed by existing benchmarks.

Building upon the limitations of current benchmarks, there is a need for more comprehensive evaluation frameworks that assess the full scope of LVLMs' capabilities. Games, with their structured environments, offer a promising solution. They inherently require not only a detailed perception of dynamic game states but also the ability to formulate strategies, anticipate an opponent's moves, and adapt to new scenarios over multiple turns. This combination of visual understanding, long-term planning, and decision-making under constraints makes games particularly well-suited for evaluating LVLMs across a wide range of cognitive and reasoning tasks. Given that even simple AI algorithms, such as Minimax and Monte Carlo Tree Search, can master games through optimization (Campbell et al., 2002), it raises the question: *Can LVLMs generalize their advanced reasoning and perception skills to perform competitively in these structured, logic-based environments?*

To address this question, we propose a game-based evaluation framework **LVLM-Playground** to assess LVLMs' abilities in structured environments thoroughly. Unlike traditional benchmarks, games provide a controlled yet dynamic setting that naturally tests perception, reasoning, decision-making and competing abilities. This framework directly addresses the key limitations of existing evaluations, as outlined below:

- **Detailed Perception.** Board games like Chess and Go require a precise perception of game states. Models must accurately interpret the positions and identities of pieces, processing fine-grained visual details essential for strategic decision-making.

- **Unique Data.** Game data is largely absent from current LVLM training sets, reducing the risk of contamination. Additionally, diverse environments and varying UI designs in games offer a wide range of novel, procedurally generated scenarios.

- **Transparent Metrics.** Games have clear, rule-based outcomes, such as winning or losing, providing objective metrics for performance evaluation and eliminating the ambiguity often found in current benchmarks that rely on human-annotated answers.

- **Consistent Prompts.** In games, the rules can serve as a uniform prompt, clearly defining what the model should do. This ensures models are evaluated on a level playing field, reducing variability and leading to more consistent, reproducible assessments.

- **Multi-turn Reasoning.** Games naturally require multi-turn interactions and long-term strategic thinking. To perform well, a model must maintain context over several moves, plan ahead, adapt dynamically to changing situations, and anticipate the opponent's moves, making games an ideal testbed for assessing sustained reasoning over extended periods.

In summary, games offer an alternative to existing benchmarks by providing a structured, dynamic, and comprehensive environment for evaluating LVLMs. In this paper, we propose **LVLM-Playground**, a game-based evaluation framework that includes six unique games: Tic-Tac-Toe, Reversi, Minesweeper, Gomoku, Sudoku, and Chess. The contributions of this work are as follows:

- We built LVLM-Playground from scratch, a comprehensive benchmark that integrates game UIs and AI opponents, enabling both online and offline interactions between LLMs and games. The

framework supports common interfaces for both commercial models, such as OpenAI API, and open-source models, like those from the HuggingFace Transformers library. LVLM-Playground includes multiple tasks with various settings, as well as automated evaluation mechanisms.

- We systematically designed a comprehensive framework to quantify the abilities required for each game and task, which includes detailed metrics that assess performance across perception, reasoning, decision-making, and adversary skills. Based on this, we evaluated state-of-the-art LVLMs, including both open-source models and commercial APIs, and generated detailed reports that highlight their strength and weaknesses under different gameplay settings.

- Based on the experimental results, we conducted an in-depth quantitative and qualitative analysis, uncovering key findings such as looping behavior in long structured outputs and poor performance in dense visual perception tasks.

## 2 RELATED WORK

### 2.1 EVALUATION FOR LVLMs

Evaluating LVLMs has proven challenging due to the need for benchmarks that comprehensively assess both language and visual modalities. Early evaluations (Liu et al., 2024a; Li et al., 2023; Alayrac et al., 2022) often focused on a limited set of vision-language tasks, with benchmarks like VQAv2 (Goyal et al., 2017), VizWiz (Bigham et al., 2010), ScienceQA (Lu et al., 2022), and Text-based VQA (Singh et al., 2019; Wang et al., 2020) being the most commonly used. Other tasks, such as image captioning (Agrawal et al., 2019; Sharma et al., 2018) and image-text retrieval (Plummer et al., 2015), were also employed to evaluate visually-conditioned language generation. However, without a unified evaluation framework, results were often reported on disparate datasets, complicating direct comparisons. In response, later works (Bai et al., 2023; Ye et al., 2024; Li et al., 2024a; Wang et al., 2024c) followed more consistent evaluation practices, relying on a few widely adopted datasets and employing radar charts to visualize model performance across tasks. This approach provided a clearer picture of each model's strengths and weaknesses, though they remained largely focused on VQA and related tasks. These benchmarks themselves were primarily designed for earlier classification-based VQA models, resulting in ground truths that are often short, lack diversity, and do not fully capture the complexities required for evaluating modern LVLMs.

To address the limitations of earlier benchmarks, recent works have developed evaluation frameworks tailored for LVLMs to capture a broader range of multimodal understanding (Wu et al., 2024; Wang et al., 2024b; Lin et al., 2024; Ji et al., 2024). For example, MMT-Bench (Ying et al., 2024) introduces 32 core tasks with 162 sub-tasks, covering areas like 3D perception and anomaly detection. Similarly, VLMEvalKit (Duan et al., 2024a) evaluates over 70 LVLMs across more than 20 benchmarks, unifying model comparisons. However, despite these expansions, many tasks still fall under VQA and its variants, limiting the diversity of benchmarked capabilities. Consequently, multiple benchmarks are often combined to assess a broader range of abilities, which increases the evaluation burden due to the large number of datasets and tasks. In comparison, the proposed LVLM-Playground offers an affordable and efficient alternative, providing comprehensive assessments of multiple capabilities without the need for exhaustive testing across hundreds of sub-tasks.

### 2.2 LVLM FOR GAMES

AI has long been applied to games, with notable achievements like Deep Blue's victory over the world chess champion (Campbell et al., 2002) and AlphaGo's mastery of Go (Silver et al., 2016), while games like Atari have also served as benchmarks for evaluating AI algorithms such as reinforcement learning (Schrittwieser et al., 2020). However, these models were trained for specific tasks with structured rules, often relying on non-visual inputs. In contrast, games with rich visual environments and complex multimodal interactions present greater challenges. Recently, LVLMs have demonstrated strong reasoning capabilities by combining visual perception with language understanding, making them well-suited to act as agents in game environments that demand both modalities for decision-making and interaction (Xu et al., 2024; Zhang et al., 2023). For example, CRADLE (Tan et al., 2024) introduces a flexible framework where LVLMs interact with games and software using screenshots and keyboard/mouse inputs, enabling them to complete complex tasks in games like *Red Dead Redemption 2 (RDR2)* without relying on built-in APIs. Similarly,

the VARP agent framework (Chen et al., 2024b) applies vision-language models to the action role-playing games like *Black Myth: Wukong*, using visual inputs for tasks such as combat and action planning. While these works focus on designing LVLM-based agent systems to perform control tasks within game environments, they do not exploit games as a systematic platform for quantitatively evaluating the models themselves. SmartPlay (Wu et al., 2023) and GAMA-Bench (Huang et al., 2024) are more closely aligned with our work, as they pioneer the use of games as a systematic benchmark for evaluating the capabilities of LLMs. However, these benchmarks primarily focus on text-based models, converting all game states into text and evaluating models in a text-only environment. In addition, there are several concurrent works that also leverage game-based environments to evaluate LLMs. GTBench (Duan et al., 2024b) explores the effects of different prompt engineering techniques, such as Chain-of-Thought and Tree-of-Thought, on the reasoning capabilities of LLMs in game scenarios, aiming to enhance their strategic decision-making processes. Similarly, GameBench (Costarelli et al., 2024) uses games as a platform to assess the strategic reasoning abilities of LLMs through various complex tasks. However, these benchmarks primarily focus on text-based models, converting all game states into textual descriptions and evaluating models in a text-only environment. As a result, they do not assess the visual perception capabilities or multimodal reasoning abilities required for real-world interactions. In comparison, the proposed LVLM-Playground provides a more comprehensive evaluation framework by leveraging both visual and textual inputs, allowing for a richer and more realistic assessment of LVLM capabilities in dynamic game environments.

## 3 LVLM-PLAYGROUND

### 3.1 OVERVIEW

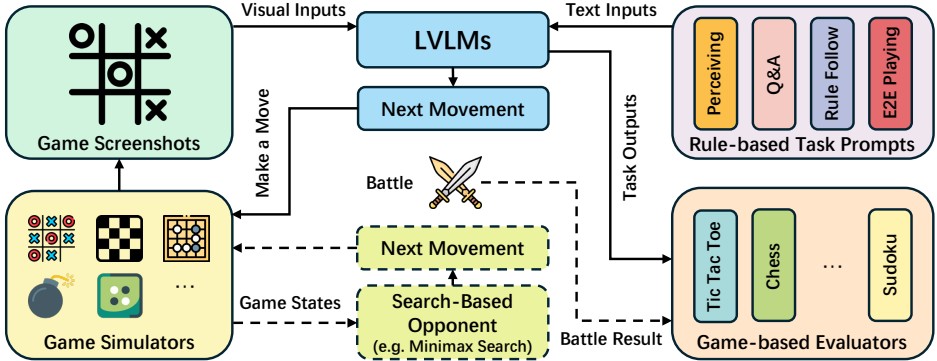

Figure 1: Overview of LVLM-Playground. LVLMs receive visual and textual inputs to perform tasks such as Perceiving, Q&A, Rule Following, and End-to-End Playing in game environments. Dashed lines indicate interactions with a search-based opponent using algorithms like Minimax or Alpha-Beta pruning in competitive games.

Previous research has found that video game playing is associated with cognitive abilities such as spatial visualization, attention control, and visual search strategies (Latham et al., 2013; Han et al., 2011). The ability of individuals, even children, to quickly grasp game mechanics and consistently apply skills related to attention, memory, and problem-solving during gameplay further reinforces this connection (Chaarani et al., 2022). Building upon these insights, LVLM-Playground leverages games to systematically evaluate the diverse cognitive and reasoning abilities of LVLMs. As illustrated in Figure 1, LVLM-Playground consists of a pool of game simulators where LVLMs interact with visual inputs, such as game screenshots, to interpret current game states. Along with task prompts based on game rules, including *Perception*, *Question Answering*, *Rule Following*, and *End-to-End Playing* (detailed in the following sections), the LVLM is expected to output either task-specific responses or the next move in the game. The selected move is then executed in the game simulator, updating the game state and generating new screenshots for further processing. In competitive games, an AI opponent is introduced, typically powered by search algorithms like Minimax

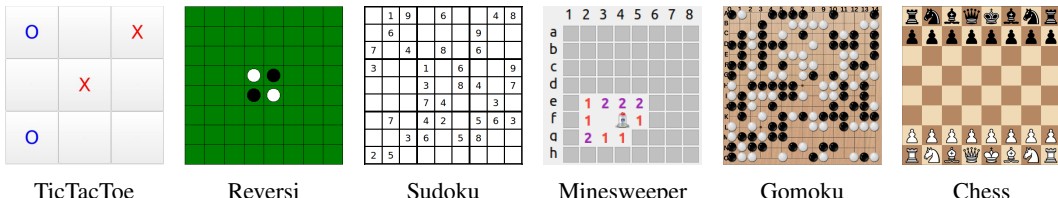

Figure 2: The LVLM-Playground comprises six different games, including Tic Tac Toe, Reversi, Sudoku, Minesweeper, Gomoku, and Chess.

to ensure a challenging adversarial environment. The results of each battle or task completion are fed into game-based evaluators, which assess performance based on specific game rules.

## 3.2 GAME SELECTION

A key consideration for building LVLM-Playground is selecting appropriate games. While recent works have explored using LVLMs in complex video games like *RDR2*, these games require low-latency, real-time actions, which current LVLMs struggle to handle due to high computational demands. As a result, their application has been largely limited to simplified scenarios, such as fragmented combat sequences, rather than full gameplay. Additionally, using such large games as benchmarks increases implementation complexity and raises the barrier to usage. Therefore, LVLM-Playground focuses on lightweight, turn-based board games that do not require strict real-time decision-making, but feature clear, well-defined rules and significant challenges that demand strong reasoning abilities. Following these principles, we selected six games (see Figure 2) that offer clear win/loss conditions or scoring systems, providing straightforward metrics for performance evaluation: **Tic Tac Toe** involves two players taking turns to place their marks on a $3 \times 3$ grid, with the objective of aligning three marks in a row, either horizontally, vertically, or diagonally. **Reversi** features players taking turns placing discs on an $8 \times 8$ board, aiming to flip the opponent's discs and control the majority of the board by the game's end. **Sudoku** requires players to fill a $9 \times 9$ grid with numbers, ensuring that each digit from 1 to 9 appears exactly once in every row, column, and $3 \times 3$ subgrid. **Minesweeper** challenges players to uncover squares on a grid, using numerical clues to avoid hidden mines while revealing safe spots. **Gomoku** is played on a $15 \times 15$ grid, where two players compete to be the first to connect five stones in a row, whether horizontally, vertically, or diagonally. **Chess** pits two players against each other on a checkered board, where each piece moves according to specific rules, with the goal of checkmating the opponent's king.

## 3.3 ABILITIES

Games have long been used in psychology to assess and enhance cognitive abilities (Von der Heiden et al., 2019; Dale & Green, 2015), with numerous studies establishing a link between gameplay and cognitive development (Choi et al., 2020; Martinez et al., 2023). Even the act of playing a simple game engages multiple cognitive processes (Boot, 2015). Players must **perceive** the game state, **reason** through rules, and make **decisions** to choose the best moves. In competitive games, players also develop **adversarial** skills by anticipating and reacting to their opponent's strategies.

Intuitively, we understand that different games present varying levels of difficulty. More complex games, like chess, require higher levels of reasoning and decision-making compared to simpler games like Tic-Tac-Toe. This is due to factors such as larger game state spaces, more intricate rules, and a greater variety of pieces and possible moves. Building on these factors, and drawing inspiration from previous work in entertainment computing (Aponte et al., 2011; Fraser et al., 2014), we developed a framework to objectively quantify the demands each game places on the four key abilities: *Perception*, *Reasoning*, *Decision*, and *Adversary*.

Taking the relatively simple game of Tic-Tac-Toe (TTT) as an illustrative example, we first review several key factors that influence the cognitive demands of the game. TTT is played on a $3 \times 3$ grid, with each cell being either marked as X, O, or left empty, resulting in three possible states per cell. Ignoring turn-by-turn rules for simplicity, the total possible game states can reach up to $3^9$. Additionally, there are two distinct piece types, X and O, and the board size is $3 \times 3$. These

Table 1: Ratings of the required abilities for each game across four key dimensions.

| Ability | Tic Tac Toe | Reversi | Sudoku | Minesweeper | Gomoku | Chess |
|---|---|---|---|---|---|---|
| Perception | ☆ | ★☆ | ★★★☆ | ★★★★ | ★★★★★ | ★★★☆ |
| Reasoning | ☆ | ★☆ | ★★ | ★★★★★ | ★★★★ | ★★★ |
| Decision | ☆ | ★★☆ | ★★ | ★★★ | ★★★☆ | ★★★★★ |
| Adversary | ☆ | ★★☆ | N/A | N/A | ★★★★ | ★★★★★ |

factors contribute to the perceptual difficulty of the game. Specifically, as the number of possible game states ($S$), piece types ($P$), and board size ($N$) increase, it becomes more challenging to accurately perceive the overall game state. This leads to the formulation for quantifying the **perception complexity** as $\Phi_{\text{perception}} = \alpha_p \log_{10}(S) + \beta_p \log_{10}(P) + \gamma_p \left[\log_{10}(N)\right]^2$, where $\alpha_p$, $\beta_p$, $\gamma_p$ are coefficients that weight the respective factors in determining the perception complexity.

Perceiving the game state is essential, but reasoning goes beyond mere observation. It involves deeper analysis, such as asking: "*How many marks has my opponent placed?*", "*Which cells are critical to control?*", "*Should I focus on defense or plan an attack?*". These considerations require players to process information and answer questions, moving beyond perception. To quantify reasoning difficulty, we consider the number of possible game states ($S$), the average branching factor ($B$), which represents the number of available moves at each stage, and the uncertainty factor ($U$). For example, in TTT, $B$ is 5, as the number of possible moves decreases from 9 as the game progresses. $U$ accounts for hidden or random information; in games like TTT, where all information is visible, $U = 0$, but in games with hidden information like Minesweeper, $U = 1$ due to uncertainty. Thus, **reasoning complexity** is quantified as $\Phi_{\text{reasoning}} = \alpha_r \log_{10}(S) + \beta_r \log_{10}(B) + \gamma_r U$, where $\alpha_r, \beta_r, \gamma_r$ are coefficients that weight the respective factors in determining the reasoning complexity.

Beyond perception and reasoning, strategic decision-making is essential, encompassing both short-term and long-term considerations. A move made early in the game can influence outcomes much later, linking decision complexity to the average game length ($L$). In TTT, for instance, games range from 5 to 9 moves, resulting in an average of 7. Additionally, decision-making involves resource management, where players must balance trade-offs, such as sacrificing one asset for a more valuable advantage. Although TTT involves managing only one type of resource (X or O), more complex games like chess require handling multiple types with varying functions, contributing to resource complexity ($C$). Finally, the average branch factor ($B$) still plays a role, as more choices at each turn add to the decision-making challenge. Therefore, we quantify **decision-making complexity** as $\Phi_{\text{decision}} = \alpha_d \log_{10}(B) + \beta_d \log_{10}(L) + \gamma_d \log_{10}(C)$, where $\alpha_d, \beta_d, \gamma_d$ are coefficients that weight the respective factors in determining the decision-making complexity.

For non-cooperative multiplayer games, the challenge extends beyond making optimal moves for oneself; it also involves anticipating and countering the opponent's strategies. Adversarial difficulty is hard to quantify, as it depends not only on the player's actions but also on the opponent's level of skill and unpredictability. To simplify this, we approximate the complexity by focusing on the possible interactions between players, where each decision is influenced by potential responses from the opponent. We quantify this by still using the average branching factor ($B$) and the average game length ($L$), but multiply them to reflect the steady increase in complexity as players react to each other's moves over the course of the game. Thus adversary complexity is approximated as $\Phi_{\text{adversary}} = L \log_{10}(B)$.

Through the above formulations, we quantify the abilities required for each game in LVLM-Playground. The resulting scores are normalized to a 0.5-5 scale, mapped to a star rating system from half a star (☆) to five stars (★★★★★), enabling comparison of the relative difficulty between games (see Table 1). Due to space constraints, please refer to Appendix A for detailed calculations and a human study validating the reasonability of the rating system.

## 3.4 TASKS

Evaluating models through E2E gameplay can leverage all the four abilities discussed in Section 3.3, yet it is unclear whether the challenges arise from issues in perceiving the game state, following the rules, or other factors. To address this, we break the evaluation into four distinct tasks: *Perceiv-*

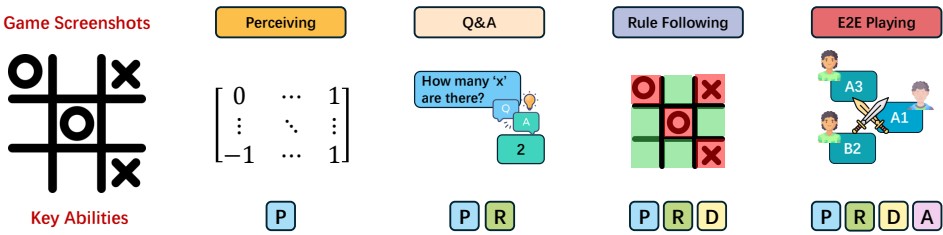

Figure 3: Models are evaluated on various games within the LVLM-Playground framework, with each game consisting of four tasks: Perceiving, Q&A, Rule Following, and E2E Playing. Each task targets one or more abilities, including **P**erception, **R**easoning, **D**ecision, and **A**dversary.

*ing*, *Question Answering*, *Rule Following*, and *E2E Playing*. Each task targets one or more of the aforementioned abilities, allowing for a more comprehensive assessment (see Figure 3).

**Perceiving** tests the model's ability to accurately capture detailed visual elements. The task requires the model to fully transcribe a randomly generated visual game state into structured representations. For example, in TTT, the model receives a screenshot of the board and must convert it into a $3 \times 3$ matrix, where 0 represents "O", 1 represents "X", and -1 denotes an empty space.

**Question Answering** extends the perceiving task by evaluating the model's ability to apply reasoning alongside perception. To align with common practices in recent Q&A-based evaluations, we adopt a multiple-choice format to help decouple instruction-following capability from perception and reasoning abilities. This approach reduces the likelihood of models being penalized for deviating from the expected output format, which allows a more focused evaluation of their core capabilities. Each game includes a set of tailored questions based on specific rules and scenarios, testing a range of abilities. For example, object counting (e.g., *"How many black pieces are on the board?"* in Gomoku), geometrical reasoning (e.g. *"Which cell has been occupied by circular marks?"* in Tic-Tac-Toe), and OCR-like tasks (e.g., *"How many mines are adjacent to the cell at row b, column 1?"* in Minesweeper). For each question, the answer is automatically generated by the game simulator, providing one correct answer along with multiple distractor options tailored to the specific game and question. Typically, four options are provided per question, although for specific question types, such as Yes/No questions, fewer options may be applied.

**Rule Following** tests the model's ability to internalize and apply game rules to identify valid moves based on the current state. For example, in Chess, pawns move forward one square, while knights move in an "L" shape. Similarly, in Reversi, a valid move must sandwich at least one of the opponent's discs between two of the player's discs. While humans can quickly grasp game rules from a few lines of instructions, models rely on task prompts to internalize these rules. In this task, a random game state is presented, and the model selects the next valid move, which is then executed in the game simulator to verify its compliance with the rules. This process also evaluates the model's in-context learning ability as it adapts to the game rules and makes informed decisions.

**End-to-End Playing** provides a comprehensive assessment of the model's competence, testing its ability to manage a game from start to finish. This task evaluates the model's overall proficiency across perception, reasoning, decision-making, and adversarial play. In competitive games like Gomoku, the model faces a search-based opponent that employs established strategies such as Minimax search, requiring the LVLMs to make strategic decisions throughout the game. In single-player games like Minesweeper, the model must independently navigate the game, making informed choices based on the current state and available information. By simulating full gameplay, this task holistically measures the model's ability to understand and succeed in complex game environments.

## 3.5 EVALUATIONS

To evaluate the models on LVLM-Playground, each task is assessed using specific metrics based on the task and game rules.

**Perceiving.** The perceiving task assesses the model's ability to convert a visual game state into a structured matrix representation. Accuracy is measured by the proportion of cor-

rectly identified elements compared to the ground truth, calculated using the formula $Acc_p = \frac{1}{m \times n} \sum_{i=1}^{m} \sum_{j=1}^{n} \mathbb{I}(P_{ij} = G_{ij})$, where $m$ and $n$ denote the matrix dimensions, $P_{ij}$ represents the model's output for cell $(i, j)$, and $G_{ij}$ is the corresponding ground truth. $\mathbb{I}(P_{ij} = G_{ij})$ equals 1 when the predicted value matches the ground truth.

**Q&A.** In the Q&A task, each question is presented in a multiple-choice format with a fixed set of candidate answers. The model's response is matched against the predefined options, and a response is considered correct if it exactly matches the ground truth answer.

**Rule Following.** The rule-following task evaluates the model's ability to apply game rules to determine valid moves. The board coordinates are specified in task prompts, allowing the LVLMs to output moves in alphanumeric formats such as A1 or B3. The game simulator then verifies whether these proposed moves are valid according to the game rules.

**E2E Playing.** The E2E playing task evaluates the model's ability to play the game from start to finish. Similar to the rule-following task, the LVLM proposes the next move in alphanumeric format, which is executed in the game simulator. If there is an opponent, the model must consider their responses; otherwise, it plays independently. To prevent indefinite loops, if the model generates three invalid moves per gameplay, it is automatically declared a loss. Since most LVLMs struggle to complete a full game, performance was evaluated based on the number of valid moves, partial game progression, and final outcomes. For example, in chess, this includes the number of valid moves made by the LVLM, the number of opponent pieces captured, and the final game result.

To provide a comprehensive comparison of model performance, we compute an aggregated score for each task by factoring in both the difficulty of each game and the specific abilities required for the task. As shown in Table 1, each game $g \in G = \{g_1, g_2, \ldots, g_n\}$ is assigned a star rating $S_{g,a}$ for each ability $a \in A$, which reflects the game's demand on that ability. In addition, Figure 3 illustrates the different combinations of abilities required by each task in $T = \{\text{Perceiving}, \text{Q\&A}, \text{Rule Following}, \text{E2E Playing}\}$. The model's performance on each game-task pair is represented by $M_{g,t}$. To aggregate these scores, we calculate the overall performance for each task using the formula $OverallScore(t) = \frac{\sum_{g \in G} S_{g,a} \cdot M_{g,t}}{\sum_{g \in G} S_{g,a}}$.

# 4 EXPERIMENTS

To validate the effectiveness of the LVLM-Playground benchmark and explore current LVLM limitations, we tested both cutting-edge commercial LVLM APIs, including GPT-4o (`gpt-4o-2024-08-06`), Gemini-1.5-Pro (`gemini-1.5-pro`), and Claude-3.5-Sonnet (`claude-3-5-sonnet-20240620`), as well as widely used open-source models like Qwen-2-VL (Wang et al., 2024a), DeepSeek-VL (Lu et al., 2024), Phi-3-VL (Abdin et al., 2024), LLaVA-1.6 (Liu et al., 2023), and Intern-VL2 (Chen et al., 2024c). All models were evaluated under the same conditions, including identical settings for maximum new tokens and task prompts. For Perceiving, Question Answering, and Rule-Following tasks, we utilized the simulator to generate 2,000 samples for each, followed by offline evaluation. For the End-to-End playing task, we conducted online evaluations, running 100 gameplays per model. In the tables, **bold** values indicate the best performance for each task, underline values highlight the best among open-source models, and blue values show performance below a random baseline.

## 4.1 RESULTS AND FINDINGS

In this section, we present detailed task and game performance of the selected LVLMs. Based on these results, we highlight several key findings that may reveal the limitations of current LVLMs.

***Finding 1. Limits in Complex Perception and Output.*** Table 2 shows LVLMs excel in simple games like Tic Tac Toe (e.g., Gemini: 0.994, Claude: 0.985), but struggle with complex boards like Gomoku and Chess. In Gomoku (15×15), models like Deepseek-vl-7b (0.010) and LLaVA-1.6-7b (0) fail to output the required matrix, often due to size errors (e.g., 15×16) or repetitive generation, while Qwen2-vl-7b (0.589) and Phi3-vl (0.593) perform better. In Chess (8×8), even top models (e.g., Claude: 0.554) show modest results, indicating challenges in perceiving dense details. This

Table 2: Quantitative results of the perceiving task across different LVLMs.

| LVLMs | TicTacToe | Reversi | Sudoku | Minesweeper | Gomoku | Chess |
|---|---|---|---|---|---|---|
| GPT-4o | 0.709 | 0.195 | 0.122 | 0.148 | 0.013 | 0.271 |
| Gemini1.5-pro | **0.994** | 0.882 | 0.723 | 0.580 | 0.583 | 0.473 |
| Claude-3.5-sonnet | 0.985 | **0.992** | **0.912** | **0.741** | **0.742** | **0.554** |
| Qwen2-vl-7b | 0.613 | 0.359 | 0.567 | 0.203 | 0.589 | 0.271 |
| Deepseek-vl-7b | 0.369 | 0.463 | 0.282 | 0.202 | 0.010 | 0.276 |
| Phi3-vl | 0.535 | 0.463 | 0.287 | 0.202 | 0.593 | 0.271 |
| LLaVA-1.6-7b | 0.302 | 0.477 | 0 | 0.188 | 0 | 0.209 |
| InternVL2-8b | 0.489 | 0.178 | 0.405 | 0.343 | 0.416 | 0.242 |
| Random | 0.332 | 0.333 | 0.103 | 0.093 | 0.332 | 0.079 |

highlights LVLMs' dual limitations: poor recognition of complex visuals and difficulty generating long structured outputs.

Table 3: Quantitative results of the Q&A task across different LVLMs.

| LVLMs | TicTacToe | Reversi | Sudoku | Minesweeper | Gomoku | Chess |
|---|---|---|---|---|---|---|
| GPT-4o | 0.812 | 0.440 | 0.430 | 0.436 | 0.351 | 0.485 |
| Gemini1.5-pro | **0.843** | 0.524 | 0.471 | 0.420 | 0.361 | 0.414 |
| Claude-3.5-sonnet | 0.821 | **0.694** | **0.694** | **0.778** | **0.467** | **0.656** |
| Qwen2-vl-7b | 0.553 | 0.409 | 0.407 | 0.546 | 0.296 | 0.441 |
| Deepseek-vl-7b | 0.326 | 0.319 | 0.260 | 0.357 | 0.265 | 0.396 |
| Phi3-vl | 0.477 | 0.354 | 0.313 | 0.470 | 0.278 | 0.281 |
| LLaVA-1.6-7b | 0.269 | 0.244 | 0.281 | 0.248 | 0.249 | 0.272 |
| InternVL2-8b | 0.600 | 0.334 | 0.365 | 0.457 | 0.280 | 0.393 |
| Random | 0.256 | 0.267 | 0.268 | 0.281 | 0.277 | 0.264 |

***Finding 2. Perception Weakness from Vision-Language Misalignment.*** Table 2 shows GPT-4o underperforms compared to commercial peers like Gemini1.5-pro and Claude-3.5-sonnet in perceiving tasks (e.g., Gomoku: 0.013 vs. 0.742, Reversi: 0.195 vs. 0.992), often hallucinating that it cannot parse images. This suggests a misalignment between its visual and language abilities, hindering complex board recognition. Similarly, LLaVA-1.6-7b struggles across tasks, scoring zero in Sudoku and Gomoku perception and falling below random baselines in Q&A. As an early model bridging vision and language with an adaptor, LLaVA's consistent poor performance highlights how such misalignment can impair both perception and reasoning in such tasks.

Table 4: Quantitative results of the rule-following task across different LVLMs.

| LVLMs | TicTacToe | Reversi | Sudoku | Minesweeper | Gomoku | Chess |
|---|---|---|---|---|---|---|
| GPT-4o | 0.895 | 0.070 | 0.462 | 0.333 | 0.413 | **0.509** |
| Gemini1.5-pro | **1.000** | 0.155 | 0.508 | **0.697** | **0.625** | 0.313 |
| Claude-3.5-sonnet | 0.874 | **0.255** | **0.721** | 0.308 | 0.506 | 0.133 |
| Qwen2-vl-7b | 0.690 | 0.174 | 0.238 | 0.570 | 0.517 | 0.440 |
| Deepseek-vl-7b | 0.521 | 0.091 | 0.212 | 0.587 | 0.509 | 0.419 |
| Phi3-vl | 0.461 | 0.145 | 0.235 | 0.567 | 0.483 | 0.370 |
| LLaVA-1.6-7b | 0.639 | 0.102 | 0.193 | 0.584 | 0.454 | 0.003 |
| InternVL2-8b | 0.706 | 0.170 | 0.206 | 0.583 | 0.510 | 0.378 |
| Random | 0.342 | 0.127 | 0.214 | 0.422 | 0.508 | 0.014 |

***Finding 3. Poor Rule Comprehension in Complex Games.*** Table 4 shows LVLMs excel in simple rule-following tasks like TicTacToe (e.g., Gemini: 1.000, GPT-4o: 0.895), but falter in complex

games like Reversi, where most models (e.g., GPT-4o: 0.070, Deepseek-vl-7b: 0.091) perform near or below the random baseline (0.127). In Gomoku, performance hovers around random (e.g., GPT-4o: 0.413, LLaVA-1.6-7b: 0.454 vs. 0.508), with Gemini (0.625) as an exception. This suggests many LVLMs fail to grasp intricate rules, often producing random moves instead of informed decisions, especially in games requiring detailed perception and strategic understanding.

Table 5: Quantitative results of the E2E task across different LVLMs.

| LVLMs | TicTacToe | Reversi | Sudoku | Minesweeper | Gomoku | Chess |
|---|---|---|---|---|---|---|
| GPT-4o | 18.8 | 94.9 | 9.2 | **82.8** | 55.5 | **70.8** |
| Gemini1.5-pro | 30.1 | **100.7** | 8.3 | 68.8 | 61.1 | 12.7 |
| Claude-3.5-sonnet | **31.0** | 70.0 | **22.5** | 68.7 | **66.7** | 32.6 |
| Qwen2-vl-7b | 20.0 | 40.0 | 1.1 | 30.2 | 10.0 | 27.9 |
| Deepseek-vl-7b | 14.9 | 40.0 | 1.5 | 36.3 | 10.0 | 10.0 |
| Phi3-vl | 20.0 | 40.0 | 1.9 | 49.8 | 30.0 | 10.0 |
| LLaVA-1.6-7b | 10.0 | 40.0 | 1.2 | 0.0 | 10.0 | 0.0 |
| InternVL2-8b | 19.2 | 40.0 | 0.9 | 38.8 | 20.0 | 10.0 |

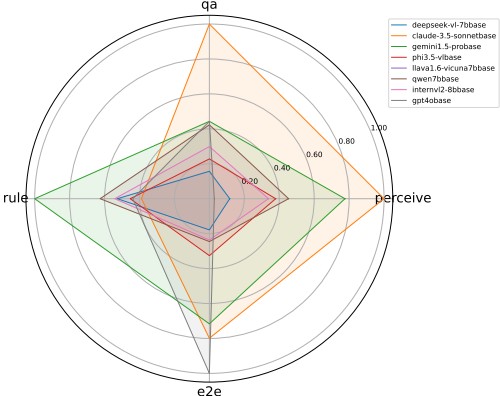

Figure 4: Game-weighted performance of LVLMs on four abilities in the LVLM-Playground.

***Finding 4. Stochastic Parrot Behavior in E2E Gameplay.*** Table 5 reveals that LVLMs struggle to complete full games in the E2E task, often exiting early due to repeated invalid moves despite scoring some intermediate points. Prompted to provide observations and strategies alongside movements, models frequently generate responses that sound plausible—describing board states or proposing plans—but fail to translate these into valid actions. For instance, in Reversi, opensource models stagnate at the initial score (40), while commercial models advance further. This pattern suggests a "stochastic parrot" behavior: LVLMs mimic gameplay language without truly understanding rules or adapting strategies, reflecting a gap between their linguistic fluency and practical game-playing ability.

***Summary.*** Figure 4 highlights LVLMs' performance across Perceiving, Q&A, Rule-Following, and E2E tasks in the LVLM-Playground benchmark. Commercial models generally excel in perception and rule comprehension, while open-source models show moderate progress but struggle with complex gameplay. Key limitations include poor handling of dense visuals, rule understanding, and sustained play, revealing gaps in vision-language integration and output coherence. This underscores the need for further advancements in LVLMs to bridge these challenges effectively.

## 5 CONCLUSION

In this paper, we present LVLM-Playground, a game-based evaluation framework for LVLMs that incorporates six diverse games across four tasks, each targeting different core abilities. LVLM-Playground addresses several key limitations of current benchmarks, offering an effective alternative for analyzing and comparing LVLMs from multiple perspectives. Using this framework, we evaluated both open-source models and commercial APIs, uncovering several key findings that highlight the limitations of existing models. In summary, LVLM-Playground provides a novel solution for evaluating the perception, reasoning, and decision-making abilities of LVLMs. We hope that our work will inspire further research and contribute new perspectives to the community.

## ACKNOWLEDGMENTS

This work was supported by the Centre for Augmented Reasoning at the Australian Institute of Machine Learning.

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

# Appendix

In this appendix, we provide additional details that were not included in the main text due to page constraints. The appendix is organized as follows:

- Reversi
- Minesweeper
- Gomoku
- Chess

# A   CALCULATION OF ABILITY RATINGS

In section 3.2, we briefly introduced the calculation of ability ratings. Here, we provide detailed steps for calculating the rating values, including how base parameters such as the number of game states ($S$), branching factor ($B$), and game length ($L$) are determined for each game.

*For simplicity, some parameters are **estimated** or **mathematically derived**, including game states that may **NOT** feasible in real gameplay.*

## A.1   WEIGHTING COEFFICIENTS

Here, we present the weighting coefficients used in the ability rating formulas in Sec 3.2.

- Perception: $\alpha_p = 0.8$, $\beta_p = 1.5$, $\gamma_p = 1.2$
- Reasoning: $\alpha_r = 1.0$, $\beta_r = 1.0$, $\gamma_r = 1.0$
- Decision: $\alpha_d = 1.0$, $\beta_d = 1.0$, $\gamma_d = 1.0$

## A.2   GAME RATING CALCULATIONS

**Tic Tac Toe.**

- **State Space Size (S) = $3^9$.** The Tic Tac Toe board is a $3 \times 3$ grid, with each cell having 3 possible states (empty, X, O). Therefore, the total state space size is $S = 3^9 = 19,683$.
- **Piece Types (P) = 2**: There are two distinct piece types in the game, X and O.
- **Board Size (N) = 9**: The board has $3 \times 3 = 9$ cells.
- **Average Branching Factor (B) = 5**: Initially, the player has 9 possible moves (since all cells are empty), and this number decreases as the game progresses. The average branching factor is approximated as $B = \frac{9+1}{2} = 5$.
- **Game Length (L) = 7**: The shortest game ends in 5 moves, while the longest game can take up to 9 moves. The average game length is approximately $G = \frac{5+9}{2} = 7$.
- **Information Uncertainty Factor (U) = 0**: Tic Tac Toe is a perfect information game with no hidden states or randomness.
- **Resource Complexity (C) = 1**: Players only manage one type of resource—their own pieces (X or O).
- **Multiplayer (is_multiplayer) = True**: Tic Tac Toe is a two-player game.

$$\Phi_P^T = 0.8 \log_{10}(3^9) + 1.5 \log_{10}(2) + 1.2 \left[\log_{10}(9)\right]^2 = 5.5057$$
$$\Phi_R^T = \log_{10}(3^9) + \log_{10}(5) + 0 = 4.9931$$
$$\Phi_D^T = \log_{10}(5) + \log_{10}(7) + \log_{10}(1) = 1.5441$$
$$\Phi_A^T = 7 \log_{10}(5) = 4.8928$$

**Minesweeper.**

- **State Space Size (S) $\approx 10^{81}$**: Minesweeper uses a $9 \times 9$ grid with 10 hidden mines. The total state space size is estimated considering all possible mine placements and grid states.
- **Piece Types (P) = 10**: There are 10 possible piece types representing different grid states: numbers 1–8 (indicating adjacent mines), mines, and unrevealed cells.

- **Board Size (N) = 81**: The board contains $9 \times 9 = 81$ cells.
- **Average Branching Factor (B) = 41**: At the start of the game, all 81 cells are unopened. The average branching factor across the game is approximately $BF = 41$.
- **Game Length (L) = 50**: On average, players perform about 50 actions in a typical game. This includes opening cells and marking mines.
- **Information Uncertainty Factor (U) = 1**: Minesweeper is a game of hidden information, as the mine locations are initially unknown to the player, making it a partially observable game.
- **Resource Complexity (C) = 1**: Players primarily manage one resource: marking or opening cells, which represents the resource complexity.
- **Multiplayer (is_multiplayer) = False**: Minesweeper is a single-player game with no interaction between players.

$$\Phi_P^M = 0.8 \log_{10}(10^{81}) + 1.5 \log_{10}(10) + 1.2 \left[\log_{10}(81)\right]^2 = 85.6423$$
$$\Phi_R^M = \log_{10}(10^{81}) + log_{10}(41) + 1 = 83.6128$$
$$\Phi_D^M = \log_{10}(41) + \log_{10}(50) + \log_{10}(1) = 3.3118$$
$$\Phi_A^M \text{ is not available.}$$

**Gomoku.**

- **State Space Size (S) = $3^{225}$**: Gomoku is played on a $15 \times 15$ grid, where each of the 225 intersections can be either empty, occupied by a black stone, or occupied by a white stone.
- **Piece Types (P) = 2**: There are two types of pieces in Gomoku: black stones and white stones.
- **Board Size (N) = 225**: The board consists of $15 \times 15 = 225$ intersections.
- **Average Branching Factor (B) = 113**: The initial move offers 225 possible actions, and the number decreases as the game progresses. The average branching factor is approximately $BF = \frac{225+1}{2} = 113$.
- **Game Length (L) = 45**: While the maximum number of moves is 225, games typically end much sooner once a player aligns five pieces in a row. The average game length is around $L = 45$.
- **Information Uncertainty Factor (U) = 0**: Gomoku is a perfect information game, meaning all information about the game state is visible to both players, with no hidden information or randomness.
- **Resource Complexity (C) = 1**: Each player manages a single type of resource, their own pieces (black or white stones).
- **Multiplayer (is_multiplayer) = True**: Gomoku is a two-player competitive game where players alternate turns, trying to outmaneuver their opponent by forming five consecutive stones in a row.

$$\Phi_P^G = 0.8 \log_{10}(3^{225}) + 1.5 \log_{10}(2) + 1.2 \left[\log_{10}(225)\right]^2 = 113.1861$$
$$\Phi_R^G = \log_{10}(3^{225}) + log_{10}(113) + 0 = 109.4054$$
$$\Phi_D^G = \log_{10}(113) + \log_{10}(45) + \log_{10}(1) = 3.7063$$
$$\Phi_A^G = 45 \log_{10}(113) = 92.3885$$

**Sudoku.**

- **State Space Size (S) $\approx 9^{81}$**: Sudoku is played on a $9 \times 9$ grid with each cell filled with a digit between 1 and 9.
- **Piece Types (P) = 9**: There are nine different types of pieces, represented by the digits 1-9.

- **Board Size (N) = 81**: The Sudoku board contains $9 \times 9 = 81$ cells.

- **Average Branching Factor (B) = 5**: While each empty cell can theoretically hold any digit from 1 to 9, the Sudoku rules limit the options. On average, a player has around 5 possible choices per empty cell.

- **Game Length (L) = 50**: Depending on the difficulty level, the game starts with some cells pre-filled. On average, a player needs to fill around 50 cells to complete the puzzle.

- **Information Uncertainty Factor (U) = 0**: Sudoku is a perfect information game, where all clues are visible on the board, and no hidden or random elements are involved.

- **Resource Complexity (C) = 1**: Players only manage one resource, the digits to be filled into the empty cells.

- **Multiplayer (is_multiplayer) = False**: Sudoku is a single-player puzzle game, where players work independently to solve the puzzle without interacting with others.

$$\Phi_P^S = 0.8 \log_{10}(9^{81}) + 1.5 \log_{10}(9) + 1.2 \left[\log_{10}(81)\right]^2 = 81.8902$$
$$\Phi_R^S = \log_{10}(9^{81}) + log_{10}(5) + 0 = 77.9926$$
$$\Phi_D^S = \log_{10}(5) + \log_{10}(50) + \log_{10}(1) = 2.3979$$
$$\Phi_A^S \text{is not available.}$$

**Chess.**

- **State Space Size (S) = $\approx 13^{64}$**: Chess is played on an $8 \times 8$ board with 64 squares, each of which can be empty or occupied by one of 12 different types of pieces (6 types, each in two colors).

- **Piece Types (P) = 12**: There are six distinct types of pieces in Chess (King, Queen, Rook, Bishop, Knight, Pawn), with each having two colors (black and white), for a total of 12 types.

- **Board Size (N) = 64**: The Chessboard contains $8 \times 8 = 64$ squares.

- **Average Branching Factor (B) = 35**: On average, a player has around 35 legal moves available per turn, though this can vary significantly depending on the game phase (opening, midgame, or endgame).

- **Game Length (L) = 80**: A typical Chess game consists of about 40 moves per player, for a total of approximately 80 moves.

- **Information Uncertainty Factor (U) = 0**: Chess is a perfect information game where all pieces and possible moves are visible to both players, and there is no element of randomness or hidden information.

- **Resource Complexity (C) = 6**: Each player manages six different types of pieces, each with unique movement rules and strategic roles, contributing to higher resource complexity.

- **Multiplayer (is_multiplayer) = True**: Chess is a two-player game, where players alternate turns, making it a multiplayer, competitive environment.

$$\Phi_P^C = 0.8 \log_{10}(13^{64}) + 1.5 \log_{10}(12) + 1.2 \left[\log_{10}(64)\right]^2 = 75.6338$$
$$\Phi_R^C = \log_{10}(13^{64}) + log_{10}(35) + 0 = 72.8364$$
$$\Phi_D^C = \log_{10}(35) + \log_{10}(80) + \log_{10}(6) = 4.2253$$
$$\Phi_A^C = 80 \log_{10}(35) = 123.5254$$

**Reversi.**

- **State Space Size (S) = $3^{64}$**: The Reversi board is an $8 \times 8$ grid with 64 squares. Each square can be empty, occupied by a black disc, or a white disc.

- **Piece Types (P) = 2**: There are two types of pieces in Reversi: black and white discs.

- **Board Size (N) = 64**: The board contains $8 \times 8 = 64$ squares.
- **Average Branching Factor (B) = 10**: Players typically have around 10 possible moves per turn on average, although this can vary throughout the game. Initially, there are fewer available moves, while mid-game offers more opportunities.
- **Game Length (L) = 60**: Reversi games generally last for about 60 moves in total, as the board has 64 squares, but not all squares are necessarily filled.
- **Information Uncertainty Factor (U) = 0**: Reversi is a perfect information game with no hidden information or randomness.
- **Resource Complexity (C) = 1**: Each player manages one type of disc (either black or white). There are no other resources to consider.
- **Multiplayer (is_multiplayer) = True**: Reversi is a two-player game, where players alternate turns placing their discs and flipping the opponent's discs.

$$\Phi_P^R = 0.8 \log_{10}(3^{64}) + 1.5 \log_{10}(2) + 1.2 \left[\log_{10}(64)\right]^2 = 34.0991$$

$$\Phi_R^R = \log_{10}(3^{64}) + log_{10}(10) + 0 = 31.5358$$

$$\Phi_D^R = \log_{10}(10) + \log_{10}(60) + \log_{10}(1) = 2.7782$$

$$\Phi_A^R = 60 \log_{10}(10) = 60.0$$

Next, we normalize the calculated ability scores to fit within a 0.5 to 5-star range. This is done by first identifying the minimum and maximum raw scores for each ability dimension across all games:

- **Perception:** $\Phi_{\min}^P = \Phi_P^T = 4.9795$, $\Phi_{\max}^P = \Phi_P^G = 113.1861$
- **Reasoning:** $\Phi_{\min}^R = \Phi_R^T = 4.9931$, $\Phi_{\max}^R = \Phi_R^G = 109.4054$
- **Decision:** $\Phi_{\min}^D = \Phi_D^T = 1.3979$, $\Phi_{\max}^D = \Phi_D^C = 4.2253$
- **Adversary:** $\Phi_{\min}^A = \Phi_A^T = 4.8928$, $\Phi_{\max}^A = \Phi_A^C = 123.5254$

Table 6 summarizes the normalized scores for each game across the four ability dimensions.

| Game | Perception | Reasoning | Decision-Making | Adversary |
|------|-----------|-----------|-----------------|-----------|
| **Tic Tac Toe** | 0.5 | 0.5 | 0.5 | 0.5 |
| **Minesweeper** | 3.86 | 5.0 | 3.08 | N/A |
| **Gomoku** | 5.0 | 4.22 | 3.57 | 3.83 |
| **Sudoku** | 3.70 | 1.82 | 1.92 | N/A |
| **Chess** | 3.45 | 2.87 | 5.0 | 5.0 |
| **Reversi** | 1.72 | 1.39 | 2.42 | 2.62 |

Table 6: Normalized ability scores for each game across the four dimensions.

We then map the normalized scores, after rounding, to the 5-star system shown in Table 1, providing a comparison of the cognitive and reasoning demands across different games and abilities.

### A.3 HUMAN STUDY OF DIFFICULTY RATINGS

To validate the reasonableness of the computed ratings, we conducted a human-based evaluation by recruiting 10 volunteers with varying levels of familiarity with the selected games. Each volunteer was asked to rank the relative difficulty of the six games based on their gameplay experiences. This ranking provided a comparative difficulty chain from the easiest to the most challenging game.

We then summed the normalized difficulty scores from Table 6 to generate an overall difficulty score for each game. For games where certain ability scores were marked as N/A, such as Minesweeper and Sudoku in the *Adversary* dimension, we assigned a minimal placeholder score of 0 to ensure consistent calculations across all games. This approach allowed us to rank the games based on their overall computed difficulty, producing a difficulty chain as follows:

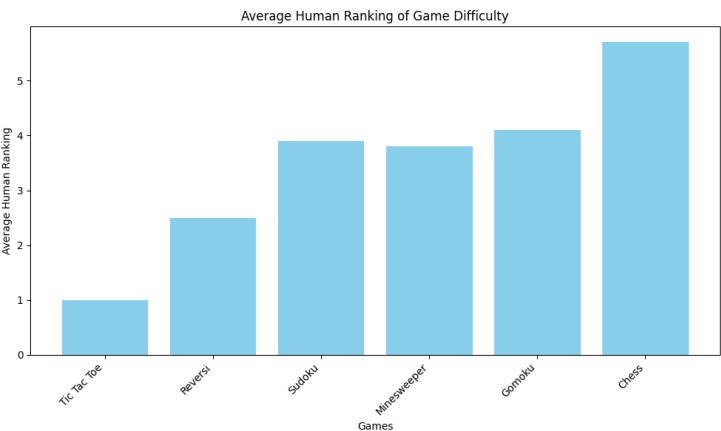

Figure 5: Average Human Ranking of Game Difficulty

*Tic Tac Toe < Reversi < Sudoku < Minesweeper < Gomoku < Chess.*

For the human-based evaluation, the rankings from all 10 volunteers were averaged to create a consolidated ranking for each game. Each volunteer's ranking was based on their perception of the overall difficulty, with the easiest game assigned a score of 1 and the most challenging game assigned a score of 6. We then calculated the average ranking for each game across all volunteers, as demonstrated in Figure 5. The comparison between the computed difficulty chain and the human-based rankings showed a high degree of consistency, supporting the reasonableness of our computed difficulty scores. We also observed that the average human rankings for Sudoku, Minesweeper, and Gomoku were very close. This suggests that, while these games differ in mechanics, they are perceived as similarly challenging. However, each of these games emphasizes different cognitive abilities which could explain the subtle differences in difficulty experienced by different players.

To further validate the evaluation settings, particularly the difficulty levels of each game across the four dimensions (perception, reasoning, decision-making, and adversary), we conducted an additional human study involving 10 volunteers.

To ensure consistency and provide the volunteers with a clear understanding of each task, we sampled 10 representative examples for each game from the LVLM-Playground simulator, covering perceiving, question-answering, and rule-following tasks. Volunteers were asked to perform these tasks themselves to gain a firsthand sense of the difficulty of each task. Additionally, to assess the adversary skills required for each game, volunteers participated in competitive matches against the search-based AI opponent in the LVLM-Playground. Since each volunteer varied in their familiarity with the games, we provided detailed game rules and guidelines beforehand to help them understand the rules and goals of each game.

Finally, we asked the volunteers to rank the four capabilities we defined for each game. The aggregated feedback is summarized as follows.

|  | Tic-Tac-Toe | Reversi | Sudoku | Minesweeper | Gomoku | Chess |
|---|---|---|---|---|---|---|
| Perception | 1.00 | 2.20 | 4.60 | 4.50 | 4.80 | 3.90 |
| Reasoning | 1.00 | 2.10 | 3.90 | 4.70 | 4.50 | 4.80 |
| Decision | 1.20 | 2.50 | 2.30 | 4.70 | 4.90 | 5.40 |
| Adversary | 1.00 | 2.50 | N/A | N/A | 2.60 | 3.90 |

As shown in the table, the human study of relative difficulty among different games demonstrates a similar trend as in the overall evaluation. Although the results do not perfectly align with the quantitative ratings of the required abilities for each game described in the paper, the trends are consistent, further validating the robustness of our evaluation framework.

# B   DETAILED GAME SETTINGS

In the main text, we provided a brief introduction to the four tasks included in the LVLM-Playground. Here, we provide more details.

## B.1   PERCEIVING

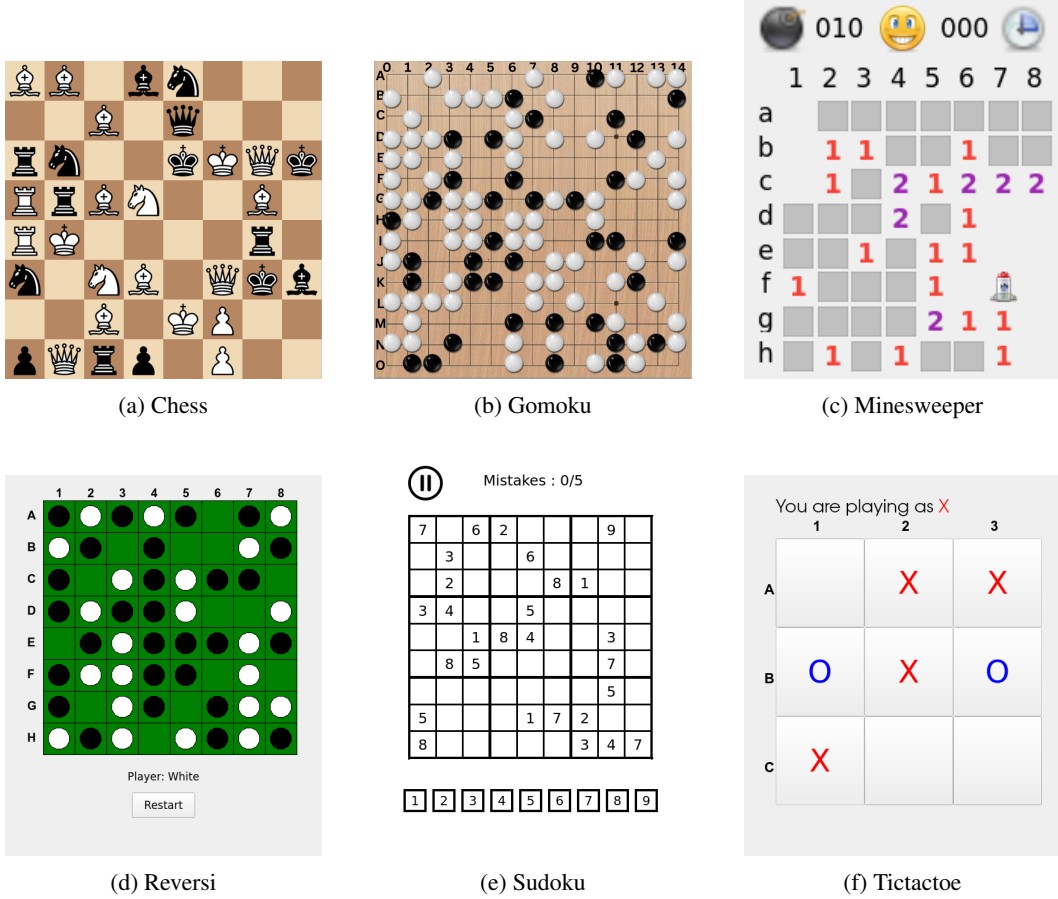

(a) Chess             (b) Gomoku             (c) Minesweeper

(d) Reversi             (e) Sudoku             (f) Tictactoe

Figure 6: Screenshots of randomly generated game states. Since these are for perception purposes only, the game rules are not guaranteed, and the game states may never occur in real games.

As shown in Figure 6, the *Perceiving* task is specifically designed to evaluate the model's ability to observe and interpret visual information from game boards, without following the actual rules of the games. The game simulator generates a wide variety of random game states, some of which may **NEVER** occur in a real gameplay scenario. For example, in Figure 6a, the chessboard shows an unrealistic scenario where there are multiple queens of the same color, which is not possible under standard chess rules. This randomness introduces complexity and variety, allowing the model to focus solely on its capacity to interpret the visual aspects of the game boards.

In general, the models are asked to convert the image game state into a structured format, such as a matrix to represent the game board. For example, in the *Reversi* game, the model is expected to output an $8 \times 8$ matrix where black pieces are represented by 1, white pieces by 2, and empty spaces by 0, accurately capturing the current game state from the visual input. Below is an example of the $8 \times 8$ matrix representing the game state shown in Figure 6d:

$$\begin{bmatrix} 1 & 2 & 1 & 2 & 1 & 0 & 1 & 2 \\ 2 & 1 & 0 & 1 & 0 & 0 & 2 & 1 \\ 1 & 0 & 2 & 1 & 2 & 1 & 1 & 0 \\ 1 & 2 & 1 & 1 & 2 & 0 & 0 & 2 \\ 0 & 1 & 2 & 1 & 1 & 1 & 2 & 1 \\ 1 & 2 & 2 & 1 & 1 & 0 & 2 & 0 \\ 1 & 0 & 2 & 1 & 0 & 1 & 2 & 2 \\ 2 & 1 & 2 & 0 & 2 & 1 & 2 & 1 \end{bmatrix}$$

The *Perceiving* task, while simple for humans, requires models to achieve high precision in local perception. For humans, interpreting a game board and translating it into structured data is a straightforward process, often done with little effort. However, for models, this task demands precise handling of the visual input and may involve capabilities like OCR to correctly interpret the elements on the board, particularly in games that include alphanumeric labels such as Sudoku or Minesweeper. Moreover, this task emphasizes the model's ability to accurately capture fine-grained details from the image and map them correctly into a structured format. Small errors in perception can lead to significant inaccuracies in the resulting matrix, which demonstrates the importance of the model's ability to perform precise local perception.

All models are provided with the same task prompt for each game, ensuring that they are operating under a fair and controlled environment. This consistency allows for direct comparison of their performance on the same task, without any variations in the instructions they receive.

Below are the task prompts provided to the models for each game:

**Perceiving Task Prompt for Chess:**

You are provided with an image of a chessboard, and your task is to represent the current state of the game as an 8x8 matrix using the specified numerical format. Each type of chess piece, both black and white, is represented by a unique number:

- Empty squares: 0
- White pieces: Pawn=1, Knight=2, Bishop=3, Rook=4, Queen=5, King=6
- Black pieces: Pawn=-1, Knight=-2, Bishop=-3, Rook=-4, Queen=-5, King=-6

From the provided chessboard image, convert the visible board into this 8x8 matrix format. For example, a typical chessboard configuration would be represented as follows:

$$\text{Game State:} \quad \begin{bmatrix} -4 & -2 & -3 & -5 & -6 & -3 & -2 & -4 \\ -1 & -1 & -1 & -1 & -1 & -1 & -1 & -1 \\ 0 & 0 & 0 & 0 & 0 & 0 & 0 & 0 \\ 0 & 0 & 0 & 0 & 0 & 0 & 0 & 0 \\ 0 & 0 & 0 & 0 & 0 & 0 & 0 & 0 \\ 0 & 0 & 0 & 0 & 0 & 0 & 0 & 0 \\ 1 & 1 & 1 & 1 & 1 & 1 & 1 & 1 \\ 4 & 2 & 3 & 5 & 6 & 3 & 2 & 4 \end{bmatrix}$$

Ensure that your output strictly follows this matrix format with no deviations, based on the pieces shown in the image.

**Perceiving Task Prompt for Gomoku: Gomoku Game State Task:**

Given a screenshot of a random 15x15 Gomoku board, please represent the current game state as a 15x15 matrix. Use 1 to represent black stones, 2 to represent white stones, and 0 to represent empty intersections. Example format:

$$
\text{Game State:} \begin{bmatrix} 0 & 0 & \dots & 0 & 0 \\ \vdots & \vdots & \ddots & \vdots & \vdots \\ 0 & 0 & 1 & \dots & 0 \\ \vdots & \vdots & \dots & 2 & \vdots \\ 0 & 0 & \dots & 0 & 0 \end{bmatrix}
$$

This represents a simplified version of a 15x15 board where 1 is a black stone, 2 is a white stone, and 0 is an empty space.

**Perceiving Task Prompt for Minesweeper:**

Minesweeper is a logic-based puzzle game played on an 8x8 grid. Each cell can either contain a mine (represented by 9), or it can be empty. Unrevealed cells should be represented by -1. Cells that are revealed and contain no adjacent mines are represented by 0, while cells that are revealed and show a number from 1 to 8 indicate how many adjacent mines surround the cell. Mines are represented by the number 9. Please strictly follow the format:

**Game State:** `<boardmatrix>`

where `<boardmatrix>` is an 8x8 matrix representing the game grid, with unrevealed cells as -1, mines as 9, and numbers from 0 to 8 indicating the number of adjacent mines. For example:

$$
\text{Game State:} \begin{bmatrix} -1 & 1 & 1 & 2 & -1 & -1 & -1 & 1 \\ 1 & 2 & 3 & 2 & -1 & 1 & 1 & 2 \\ 2 & 3 & 4 & 3 & 1 & 1 & 1 & 2 \\ 1 & 2 & 2 & 2 & 2 & 2 & 2 & 1 \\ 1 & 2 & 2 & 2 & 2 & 9 & 1 & 0 \\ -1 & 1 & 2 & 3 & 2 & 1 & 0 & -1 \\ -1 & -1 & 1 & 2 & 3 & 1 & -1 & -1 \\ 1 & 2 & 9 & 1 & 1 & -1 & -1 & -1 \end{bmatrix}
$$

This example represents a grid where some cells have been uncovered, showing numbers indicating adjacent mines, unrevealed cells are represented by -1, and mines are represented by the number 9.

**Perceiving Task Prompt for Reversi:**

Reversi (also known as Othello) is a strategy board game played on an 8x8 grid, where two players take turns placing black and white pieces on the board. The goal is to have more pieces of your color on the board than your opponent by the end of the game. A piece is placed on an empty square and must sandwich one or more of the opponent's pieces between the newly placed piece and another of the player's pieces in a horizontal, vertical, or diagonal line. The opponent's pieces in between are then flipped to the player's color. Given a screenshot of the Reversi board, please represent the current state of the game using an 8x8 matrix. In this matrix, empty cells should be represented by 0, black pieces by 1, and white pieces by 2. Please strictly follow the format:

**Game State:** `<boardmatrix>`

where `<boardmatrix>` is an 8x8 matrix. For example:

$$
\text{Game State:} \begin{bmatrix}
0 & 0 & 0 & 0 & 0 & 0 & 0 & 0 \\
0 & 0 & 0 & 0 & 0 & 0 & 0 & 0 \\
0 & 0 & 0 & 0 & 0 & 0 & 0 & 0 \\
0 & 0 & 0 & 2 & 1 & 0 & 0 & 0 \\
0 & 0 & 0 & 1 & 2 & 0 & 0 & 0 \\
0 & 0 & 0 & 0 & 0 & 0 & 0 & 0 \\
0 & 0 & 0 & 0 & 0 & 0 & 0 & 0 \\
0 & 0 & 0 & 0 & 0 & 0 & 0 & 0
\end{bmatrix}
$$

represents a Reversi board with a few pieces already placed and empty cells (represented by 0).

**Perceiving Task Prompt for Sudoku:**

Sudoku is a logic-based puzzle played on a 9x9 grid, where the grid is subdivided into nine 3x3 subgrids. The goal is to fill the grid so that each row, each column, and each 3x3 subgrid contains all digits from 1 to 9 without repetition. Given a screenshot of the Sudoku grid, please represent the current state of the puzzle using a 9x9 matrix. In this matrix, an empty cell should be represented by 0, and filled cells should contain their respective numbers (1-9). Please strictly follow the format:

**Game State:** `<boardmatrix>`

where `<boardmatrix>` is a 9x9 matrix. For example:

$$
\text{Game State:} \begin{bmatrix}
5 & 3 & 0 & 0 & 7 & 0 & 0 & 0 & 0 \\
6 & 0 & 0 & 1 & 9 & 5 & 0 & 0 & 0 \\
0 & 9 & 8 & 0 & 0 & 0 & 0 & 6 & 0 \\
8 & 0 & 0 & 0 & 6 & 0 & 0 & 0 & 3 \\
4 & 0 & 0 & 8 & 0 & 3 & 0 & 0 & 1 \\
7 & 0 & 0 & 0 & 2 & 0 & 0 & 0 & 6 \\
0 & 6 & 0 & 0 & 0 & 0 & 2 & 8 & 0 \\
0 & 0 & 0 & 4 & 1 & 9 & 0 & 0 & 5 \\
0 & 0 & 0 & 0 & 8 & 0 & 0 & 7 & 9
\end{bmatrix}
$$

represents a partially filled Sudoku grid with some cells empty (represented by 0).

**Perceiving Task Prompt for Tic Tac Toe:**

Tic Tac Toe is a game played on a 3x3 grid where players take turns placing X or O in the cells. Given a screenshot of the game board, please determine the current game state using a 3x3 matrix. In this matrix, an empty cell should be represented by -1, X should be represented by 1, and O should be represented by 0. Please strictly follow the format:

**Game State:** `<boardmatrix>`

where `<boardmatrix>` is a 3x3 matrix. For example:

$$\text{Game State: } \begin{bmatrix} -1 & -1 & -1 \\ -1 & -1 & -1 \\ -1 & -1 & -1 \end{bmatrix}$$

represents an empty board.

As shown in the prompts above, each task specifies a clear and structured output format that the models are expected to follow. For each game, the models must generate a matrix representing the current state of the game board. These matrices must conform to the exact specifications outlined in the task prompts, ensuring that the output is both precise and standardized.

To evaluate the model's performance on the *Perceiving* task, we compute the accuracy by comparing the model's predicted matrix with the ground truth matrix. Each task specifies a clear expected output format, typically a matrix representing the current game state. Let $P$ represent the predicted matrix and $G$ represent the ground truth matrix, both of size $m \times n$, where $m$ and $n$ are the dimensions of the specific game board. The accuracy can be calculated using the following formula:

$$Acc_p = \frac{1}{m \times n} \sum_{i=1}^{m} \sum_{j=1}^{n} \mathbb{I}(P_{ij} = G_{ij})$$

To ensure the responses are in the correct format, we apply a regex to extract the matrix from the model's output. If the model's response cannot be parsed by the regex or does not adhere to the expected structure, it is marked as invalid, and a zero accuracy score is assigned for that task. This approach ensures a fair and consistent evaluation across all models.

## B.2 QUESTION ANSWERING

The *Question Answering* task follows the structure of traditional VQA but is adapted to a game-based environment. A key advantage of this setup is that specific game states can be easily obtained from the simulator, allowing for the automatic generation of question-answer pairs without manual annotation. This contrasts with traditional VQA, which requires significant human effort for data collection and labeling. Additionally, this approach enables detailed manipulation of the input, allowing us to evaluate models on **fine-grained perception** and reasoning based on visual details. Unlike standard VQA, which often focuses on broader scene-level questions, LVLM-Playground encourages models to engage with and reason about visual details.

We generate screenshots of randomly created game states, similar to the process used in the perceiving task (as shown in Figure 6). For each game, a specific question pool is designed, accompanied by APIs to retrieve the corresponding ground truth answers. These questions are crafted to assess the model's understanding of the game state and its reasoning based on visual input.

To address the common challenge in VQA evaluation of **variability in response format** from language models, we adopt a **multiple-choice** format for the Q&A task. Each question is paired with a predefined set of candidate answers, including one correct answer and several distractors. This setting eliminates the reliance on free-form responses and simplifies the validation process, focusing the evaluation on the model's reasoning and perception abilities rather than instruction-following.

For each game, questions are designed with corresponding candidate answers, typically containing four options, although fewer options may be provided for binary questions (e.g., Yes/No). The

multiple-choice format ensures consistency across tasks and enables direct comparison of model performance. The prompt structure includes clear instructions and examples of valid question-answer pairs to guide the model in selecting the correct answer.

**Q&A Task Prompt for Chess:**

> Chess is played on an 8x8 board with six types of pieces, including pawns, knights, bishops, rooks, queens, and kings, for both white and black players. The board uses a coordinate system where columns are labeled "a" through "h" from left to right, and rows are labeled "1" through "8" from bottom to top. For example, the bottom-left corner is "a1" and the top-right corner is "h8". Please answer the following question based on the provided screenshot of the current game state:
> {question}
> **Answer:** <answer>
> where <answer> should be one of A, B, C, or D.

**Q&A Task Prompt for Gomoku:**

> Gomoku is played on a 15x15 grid, where black and white stones are placed in turns. The goal is to place five consecutive stones in a horizontal, vertical, or diagonal line. Please answer the following question based on the provided screenshot of the current game state:
> {question}
> **Answer:** <answer>
> where <answer> should be one of A, B, C, or D.

**Q&A Task Prompt for Minesweeper:**

> Minesweeper is played on an 8x8 grid with exactly 10 mines. Each cell can either contain a mine, be unrevealed, or show the number of adjacent mines (from 0 to 8). The grid is labeled with rows A to H (top to bottom) and columns 1 to 8 (left to right). Please answer the following question based on the provided screenshot of the current game state:
> {question}
> **Answer:** <answer>
> where <answer> should be one of A, B, C, or D.

**Q&A Task Prompt for Reversi:**

> Reversi (also known as Othello) is played on an 8x8 grid where two players take turns placing black and white pieces on the board. Please answer the following question based on the provided screenshot of the current game state:
> {question}
> **Answer:** <answer>
> where <answer> should be one of A, B, C, or D.

**Q&A Task Prompt for Sudoku:**

> Sudoku is played on a 9x9 grid, where each row, column, and 3x3 subgrid must contain the numbers 1 to 9 exactly once. Please answer the following question based on the provided screenshot of the current game state:
> {question}
> **Answer:** \<answer\>
> where \<answer\> should be one of A, B, C, or D.

**Q&A Task Prompt for Tic Tac Toe:**

> Tic Tac Toe is a game played on a 3x3 grid where two players take turns placing X or O in the cells. The goal is to form a horizontal, vertical, or diagonal line with three of your own marks. The grid is labeled with rows A to C and columns 1 to 3. Please answer the following question based on the provided screenshot of the current game state:
> {question}
> **Answer:** \<answer\>
> where \<answer\> should be one of A, B, C, or D.

As shown above, only responses that adhere to the specified format and contain the correct answer will contribute to the accuracy. Otherwise, even if the answer is correct but presented in an unexpected format, it might be marked as incorrect. This is reasonable, much like in human exams where failure to follow formatting rules can result in penalties, regardless of the correctness of the answer. Therefore, the ability to strictly follow instructions is crucial for the model, as it reflects whether it truly understands the given instructions.

For each game, we have designed a unique set of questions tailored to its specific dynamics and rules. Below, we outline the types of questions used in each game and how they test different aspects of the model's perception and reasoning capabilities.

**Chess Question Types.**

- **Piece Color at Position**
  *Question*: What is the color of the piece at column {col_label}, row {row_label}?

- **Piece Name at Position**
  *Question*: What is the piece at column {col_label}, row {row_label}?

- **Count of Specific Pieces**
  *Question*: How many {piece_color} {piece_name}s are on the board?

- **Count of Pieces in Row/Column**
  *Question*: How many pieces are in row {row_label}?
  OR
  *Question*: How many pieces are in column {col_label}?

- **Comparison of White and Black Pieces**
  *Question*: Which color has more pieces, white or black?

- **Comparison of Two Piece Types**
  *Question*: Which has more, {piece1_name}s or {piece2_name}s?

- **Count of Edge Pieces**
  *Question*: How many pieces are on the edge of the board?

- **More Empty Cells in Top or Bottom Half of the Board**
  *Question*: Which half of the board has more empty cells, top or bottom?

**Gomoku Question Types.**

- **Stone at Specific Position**
  *Question*: What is the stone at row {row_label}, column {col_idx}?

- **Count of Specific Stones**
  *Question*: How many 'Black' stones are on the board?
  OR
  *Question*: How many 'White' stones are on the board?

- **Count of Stones in Row/Column**
  *Question*: How many 'Black' stones are in row {row_label}?
  OR
  *Question*: How many 'White' stones are in column {col_idx}?

- **Winning Condition**
  *Question*: Is there a winning line on the board?

- **Adjacent Stones**
  *Question*: How many adjacent stones are around row {row_label}, column {col_idx}?

- **Count of Empty Cells**
  *Question*: How many empty cells are there on the board?

- **Maximum Consecutive Stones**
  *Question*: What is the maximum number of consecutive 'Black' stones in row {row_label}?
  OR
  *Question*: What is the maximum number of consecutive 'White' stones on any diagonal?

- **Count of Edge Stones**
  *Question*: How many 'Black' stones are on the edge of the board?
  OR
  *Question*: How many 'White' stones are on the edge of the board?

**Minesweeper Question Types.**

- **Revealed Symbol at Specific Position**
  *Question*: What is the revealed number or symbol in row {row_label}, column {col_label}?

- **Count of Revealed Mines**
  *Question*: How many revealed mines are there on the board?

- **Count of Revealed Cells**
  *Question*: How many revealed cells are there on the board?

- **Count of Revealed Cells in Row/Column**
  *Question*: How many revealed cells are there in row {row_label}?
  OR
  *Question*: How many revealed cells are there in column {col_label}?

- **Number of Adjacent Mines**
  *Question*: How many mines are adjacent to the cell at row {row_label}, column {col_label}?

- **Is There a Mine at a Specific Position**
  *Question*: Is there a revealed mine at row {row_label}, column {col_label}?

**Reversi Question Types.**

- **Symbol at Specific Position**
  *Question*: What is the symbol in row {row_label}, column {col_label}?

- **Count of Specific Symbol**
  *Question*: How many 'Black' pieces are present on the board?
  OR
  *Question*: How many 'White' pieces are present on the board?

- **Count of Empty Cells**
  *Question*: How many empty cells are there on the board?

- **Count of Specific Symbol in Row/Column**
  *Question*: How many 'Black' pieces are there in row {row_label}?
  OR
  *Question*: How many 'White' pieces are there in column {col_label}?

- **Row with Most Pieces of Specific Symbol**
  *Question*: Which row contains the most 'Black' pieces?
  OR
  *Question*: Which row contains the most 'White' pieces?

- **Comparison of Black and White Pieces**
  *Question*: Which player has more pieces on the board, 'Black' or 'White'?

- **Total Number of Pieces in Row/Column**
  *Question*: How many pieces are there in total in row {row_label}?
  OR
  *Question*: How many pieces are there in total in column {col_label}?

- **Total Count of Black or White Pieces on the Board**
  *Question*: How many 'Black' pieces are there in total on the board?
  OR
  *Question*: How many 'White' pieces are there in total on the board?

**Sudoku Question Types.**

- **Symbol at Specific Position**
  *Question*: What is the number in row {row}, column {col}?

- **Count of Specific Number**
  *Question*: How many '{number}'s are present on the board?

- **Count of Empty Cells**
  *Question*: How many empty cells are there on the board?

- **Count of Specific Number in Row/Column**
  *Question*: How many '{number}'s are there in row {row}?
  OR
  *Question*: How many '{number}'s are there in column {col}?

- **Count of Specific Number in Subgrid**
  *Question*: How many '{number}'s are there in the subgrid starting at row {subgrid_start_row}, column {subgrid_start_col}?

- **Sum of Numbers in Row/Column**
  *Question*: What is the sum of numbers in row {row}?
  OR
  *Question*: What is the sum of numbers in column {col}?

- **Sum of Numbers in Subgrid**
  *Question*: What is the sum of numbers in the subgrid starting at row {subgrid_start_row}, column {subgrid_start_col}?

- **Does Row/Column Contain a Specific Number**
  *Question*: Does row {row} contain the number '{number}'?
  OR
  *Question*: Does column {col} contain the number '{number}'?

- **Count of Empty Cells in Subgrid**
  *Question*: How many empty cells are there in the subgrid starting at row {subgrid_start_row}, column {subgrid_start_col}?

**Tic Tac Toe Question Types.**

- **Symbol at Specific Position**
  *Question*: What is the symbol in row {row}, column {col}?

- **Count of Specific Symbol**
  *Question*: How many '{symbol}'s are present on the board?

- **Count of Empty Cells**
  *Question*: How many empty cells are there?

- **Count of Specific Symbol in Row/Column**
  *Question*: How many '{symbol}'s are there in row {row}?
  OR
  *Question*: How many '{symbol}'s are there in column {col}?

- **Winner of the Game**
  *Question*: Did X or O win the game?

- **Count of Red/Blue Marks on the Board**
  *Question*: How many {color} marks are present on the board?
  OR
  *Question*: How many {color} marks are there in row {row}?
  OR
  *Question*: How many {color} marks are there in column {col}?

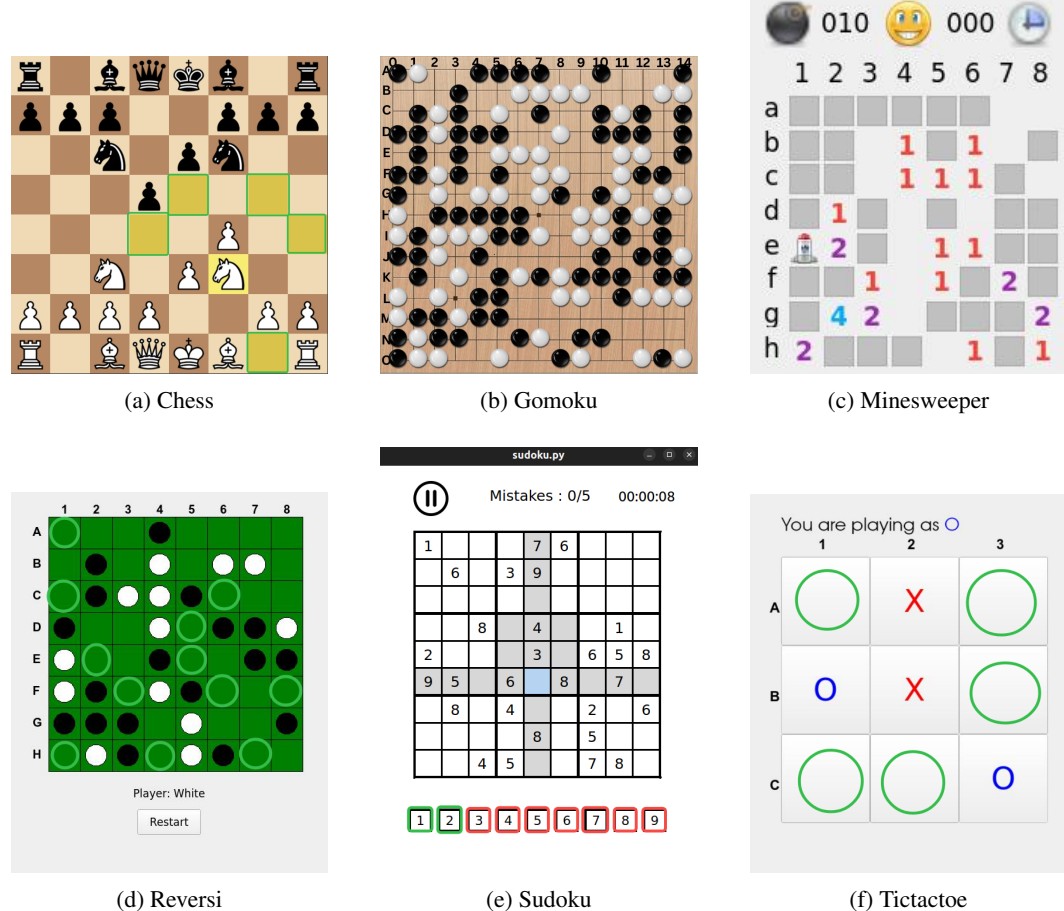

Figure 7: Visualization of the Rule Following task, where the model is presented with a random game state and required to select a valid move. In (a) Chess, (d) Reversi, (e) Sudoku, and (f) TicTacToe, valid moves are highlighted in green rectangles or circles. In (b) Gomoku, valid moves are represented by the empty intersections, and in (c) Minesweeper, valid moves correspond to the uncovered (gray) cells.

As shown in the examples above, the question-answering task challenges the model's ability to engage with various question types that require reasoning and fine-grained perception. Unlike the pure visual *perceiving* task, these questions go further by testing the model's reasoning skills and multiple detailed analytical abilities. For example, some questions involve **counting**, such as determining how many pieces of a certain type are present on a board, while others require basic **arithmetic calculations**, such as summing numbers in rows or columns in Sudoku. There are also tasks that require **OCR-like recognition**, identifying symbols or numbers at specific positions, as well as questions that demand **geometrical shape recognition**, such as identifying edge pieces or distinguishing between shapes like circles and rectangles. These capabilities, which usually require separate benchmarks to evaluate, are integrated within the proposed LVLM-Playground, offering a comprehensive assessment of the model's ability to perceive, reason, and handle fine-grained visual details.

## B.3 RULE FOLLOWING

The Rule Following task evaluates the model's ability to internalize and execute game rules accurately, assessing whether the model can determine and perform valid moves in accordance with the specific rules of each game. Unlike the Perceiving and Q&A tasks, where the randomly generated game states may not strictly follow the game rules, the Rule Following task ensures that all randomly

generated game states adhere to the rules of the game. Then, the model is tasked with selecting a valid move based on the state.

In chess (Figure 7a), each piece has specific movement rules, such as pawns moving one square forward. These moves can be described using algebraic notation, where each square is identified by a letter and a number. For instance, a valid pawn move might be from *e2* to *e4*, indicating that the pawn moves two squares forward from its starting position.

In Gomoku (Figure 7b) and Minesweeper (Figure 7c), valid moves are selecting empty intersections on the board or unmarked (gray) cells on the Minesweeper map.

In Reversi (Figure 7d), a valid move must sandwich at least one of the opponent's discs between two of the player's discs.

In Sudoku (Figure 7e), a number must not be repeated within the same row, column, or subgrid.

In Tic-Tac-Toe (Figure 7f), a move is valid if it occupies an empty cell, as each cell can only be filled once.

For each game, we provide detailed game rules through structured prompts to test the model's ability to internalize and apply them. Specifically, the rule-following prompts for each game are as follows:

**Rule Following Task Prompt for Chess:**

> Chess is played on an 8x8 board following standard chess rules. Each piece moves according to its unique capabilities. The board uses algebraic coordinates with files labeled `a` through `h` from left to right and ranks labeled `1` through `8` from bottom to top (a1 at bottom-left, h8 at top-right). White moves first, followed by Black. Based on the current board state image, please choose one legal move for White and output it using Standard Algebraic Notation (SAN). For example, if White's pawn on `e2` can move to `e4`, your answer should be `e4`; if White's knight on `g1` can move to `f3`, your answer should be `Nf3`.
>
> Please strictly follow the following format:
>
> **Movement:** `<move>`
>
> where `<move>` is the move in SAN (e.g., `e4`, `Nf3`, `O-O`).

**Rule Following Task Prompt for Gomoku:**

> Gomoku is played on a 15x15 grid, where black and white stones are placed on the intersections of the grid. The objective is to place five consecutive stones in a horizontal, vertical, or diagonal line. The game starts with an empty board. The grid is labeled with columns A to O (left to right) and rows 1 to 15 (top to bottom). Each intersection can only be occupied by one stone, either black or white. Based on the board state, please find an empty intersection where you can place your next stone.
>
> Please strictly follow the following format:
>
> **Movement:** `<position>`
>
> where the `<position>` can be any combination of columns A to O and rows 1 to 15, for example, A1, H8, or O15.

**Rule Following Task Prompt for Minesweeper:**

Minesweeper is played on an 8x8 grid with exactly 10 mines. Each cell can either contain a mine, be unrevealed, or show the number of adjacent mines (from 0 to 8). The goal is to reveal all cells that do not contain mines, without triggering any mines. You win when only the 10 mine cells remain unrevealed. The grid is labeled with rows A to H (top to bottom) and columns 1 to 8 (left to right). Each cell can only be revealed once, and flagging is not allowed. Based on the current board state image, please find a cell to reveal next.

Please strictly follow the following format:

**Movement:** `<position>`

where the `<position>` can be any combination of rows A to H and columns 1 to 8, for example, A1, B5, or H8.

**Rule Following Task Prompt for Reversi (Othello):**

Reversi (also known as Othello) is played on an 8x8 grid. Players take turns placing black and white pieces on the board. The grid starts with two black pieces and two white pieces in the center (`D4`, `D5`, `E4`, `E5`). A valid move consists of placing a piece in such a way that it sandwiches one or more of the opponent's pieces between the newly placed piece and another of the player's pieces in a horizontal, vertical, or diagonal line. After placing the piece, all of the opponent's pieces in between are flipped to the player's color. The black player (you) goes first, followed by the white player (AI). The grid is labeled with rows A to H and columns 1 to 8. Based on the current game state, please find a valid position where you can place your next black piece and flip at least one of the opponent's white pieces.

Please strictly follow the format:

**Movement:** `<position>`

where the `<position>` can be any combination of rows A to H and columns 1 to 8, for example, `A1`, `D4`, or `H8`.

**Rule Following Task Prompt for Sudoku:**

Sudoku is played on a 9x9 grid, where each row, column, and 3x3 subgrid must contain the numbers 1 to 9 exactly once. The grid starts with the top-left corner as A1, where rows are labeled from A to I and columns are numbered from 1 to 9. A valid move involves placing a digit from 1 to 9 in an empty cell, ensuring that the number does not already appear in the same row, column, or 3x3 subgrid. Based on the current state of the Sudoku grid, please find a valid empty cell where you can place a digit and make a valid move.

Please strictly follow the format:

**Movement:** `<row><column> <digit>`

where `<row>` is A to I (representing rows), `<column>` is 1 to 9 (representing columns), and `<digit>` is a number between 1 to 9. For example, `A1 5` means placing the digit 5 in the top-left corner of the grid.

**Rule Following Task Prompt for Tic Tac Toe:**

> Tic Tac Toe is a game played on a 3x3 grid where two players take turns placing their respective marks, X or O, in the cells. The game begins with an empty board, and the grid is labeled with rows A to C and columns 1 to 3. Each position on the grid can only be occupied by one mark at a time. Based on the current state of the board, your task is to find an empty cell where you can place your next stone. Please follow the format below when indicating your move:
>
> **Movement:** `<position>`
>
> The position should be a combination of a row (A to C) and a column (1 to 3), such as A1, B2, or C3.

## B.4    END-TO-END PLAYING

E2E Playing task evaluates the model's capability to autonomously play a game from start to finish, testing its ability to integrate perception, reasoning, and decision-making. As discussed in the main text, the model suggests the next move in alphanumeric format, which is then executed within a game simulator. For games involving an opponent, the model must respond to the adversarial actions and adapt its strategy accordingly.

Here, we first provide the E2E Play task prompt for each game:

**End-to-End Playing Task Prompt for Tic Tac Toe:**

> Tic Tac Toe is played on a 3x3 grid. Players take turns placing X or O in the cells. The goal is to be the first to form an unbroken line of three marks horizontally, vertically, or diagonally. The game starts with an empty board, and the O player goes first. The grid is labeled with rows A to C and columns 1 to 3. You are playing as O, aiming to win by placing marks strategically. Each position can only be occupied by one mark, so do not choose a spot that is already taken. Based on the board state screenshots, please first observe the current situation, then carefully think and explain your strategy briefly, and finally output your movement for this status.
>
> Please strictly follow the following format:
>
> **Observation:** `<observation>`
> **Strategy:** `<strategy>`
> **Movement:** `<position>`
>
> Where the observation should briefly summarize the current situation, the strategy is a brief explanation of how you plan to win the game, and the position can be any combination of rows A to C and columns 1 to 3, for example, A1, 2B, or C3.

**End-to-End Playing Task Prompt for Sudoku:**

Sudoku is a logic-based puzzle played on a 9x9 grid, subdivided into nine 3x3 subgrids. The goal is to fill the grid so that each row, column, and 3x3 subgrid contains all digits from 1 to 9 without repetition. The grid starts with some cells filled with numbers (clues), and you must fill in the remaining empty cells one at a time. Rows are labeled A to I (top to bottom), and columns are numbered 1 to 9 (left to right). Each move involves placing a digit (1-9) in an empty cell, ensuring no repetition in its row, column, or 3x3 subgrid. Based on the current board state screenshot, observe the situation, formulate a strategy, and output a single valid move to place a digit.

Please strictly follow the format:

**Observation:** `<observation>`
**Strategy:** `<strategy>`
**Movement:** `<row> <column> <digit>`

Where `<row>` is A to I, `<column>` is 1 to 9, and `<digit>` is 1 to 9. For example, `A1 5` places 5 in the top-left cell.

**End-to-End Playing Task Prompt for Reversi (Othello):**

Reversi (also known as Othello) is a strategy board game played on an 8x8 grid. Players take turns placing black and white pieces on the board. The goal is to have more pieces of your color on the board than your opponent by the end of the game. A valid move must sandwich one or more of the opponent's pieces between the newly placed piece and another of your pieces in a horizontal, vertical, or diagonal line, flipping the sandwiched pieces to your color.

The game starts with four pieces in the center: two black (at `D4` and `E5`) and two white (at `D5` and `E4`). The black player (you) goes first, followed by the white player (AI). The grid is labeled with rows A to H (top to bottom) and columns 1 to 8 (left to right). You are playing as black, aiming to maximize your pieces by placing them strategically.

Based on the board state screenshots, please first observe the current situation, then carefully think and explain your strategy briefly, and finally output your movement for this status.

Please strictly follow the format:

**Observation:** `<observation>`
**Strategy:** `<strategy>`
**Movement:** `<position>`

Where the observation should briefly summarize the current situation, the strategy is a brief explanation of how you plan to maximize your pieces, and the position can be any combination of rows A to H and columns 1 to 8, for example, `A1`, `D4`, or `H8`.

**End-to-End Playing Task Prompt for Minesweeper:**

Minesweeper is a logic-based puzzle game played on an 8x8 grid with exactly 10 mines. Each cell can either contain a mine, be unrevealed, or show the number of adjacent mines (from 0 to 8). The goal is to reveal all cells that do not contain mines without triggering any mines. You win when only the 10 mine cells remain unrevealed.

The grid is labeled with rows A to H (top to bottom) and columns 1 to 8 (left to right). You can only reveal cells (e.g., `A1`), and flagging is not allowed.

Based on the current board state screenshot, please observe the situation, formulate a strategy, and output a move to reveal a cell.

Please strictly follow the format:

**Observation:** `<observation>`
**Strategy:** `<strategy>`
**Movement:** `<row>` `<column>`

Where `<observation>` summarizes the current board state, `<strategy>` explains your reasoning, and `<position>` is a move in the format `A1`.

**End-to-End Playing Task Prompt for Gomoku:**

Gomoku is played on a 15x15 grid, where players take turns placing black or white stones on the intersections. The goal is to be the first to form an unbroken line of five stones horizontally, vertically, or diagonally.

The game starts with an empty board, and the black player (you) goes first, followed by the white player (AI). The grid is labeled with columns A to O (left to right) and rows 1 to 15 (top to bottom). You are playing as black, aiming to win by placing stones strategically. Each intersection can only be occupied by one stone, so do not choose a spot that is already taken.

Based on the board state screenshots, please first observe the current situation, then carefully think and explain your strategy briefly, and finally output your movement for this status.

Please strictly follow the format:

**Observation:** `<observation>`
**Strategy:** `<strategy>`
**Movement:** `<position>`

Where the observation should briefly summarize the current situation, the strategy is a brief explanation of how you plan to win the game, and the position can be any combination of columns A to O and rows 1 to 15, for example, `A1`, `H8`, or `O15`.

**End-to-End Playing Task Prompt for Chess:**

> Chess is a strategy game played on an 8x8 board with 64 squares, using six types of pieces: pawns, knights, bishops, rooks, queens, and kings, for both white and black players. The game starts with a standard initial position: white pieces on ranks 1 and 2, black pieces on ranks 7 and 8. The board uses algebraic coordinates with files labeled a through h from left to right and ranks labeled 1 through 8 from bottom to top (`a1` at bottom-left, `h8` at top-right). White moves first, followed by Black.
>
> You are playing as White, aiming to checkmate the Black king or achieve a favorable position. Each move must follow standard chess rules and be expressed in Standard Algebraic Notation (SAN), such as `e4` (pawn to `e4`), `Nf3` (knight to `f3`), or `O-O` (kingside castling).
>
> Based on the board state screenshots, please first observe the current situation, then carefully think and explain your strategy briefly, and finally output your movement for this status.
>
> Please strictly follow the following format:
>
> **Observation:** `<observation>`
> **Strategy:** `<strategy>`
> **Movement:** `<algebraic notation>`
>
> Where the observation should briefly summarize the current situation, the strategy is a brief explanation of how you plan to win or improve your position, and the position is a legal move in SAN, for example, `e4`, `Nf3`, or `O-O`.

The scoring rules for the E2E Playing task are designed to reflect the model's performance in terms of efficiency, accuracy, and game outcome across different games. Below, we describe the scoring mechanism for each game based on its implementation:

- **Tic Tac Toe:** The score combines a base score, calculated as 10 points per move made by the player (O), and an outcome bonus: 50 points for a win, 20 points for a tie, and 0 points for a loss. The total score is:

$$\text{Score} = (N_{\text{moves}} \times 10) + B_{\text{outcome}}, \quad B_{\text{outcome}} = \begin{cases} 50 & \text{if win} \\ 20 & \text{if tie} \\ 0 & \text{if loss} \end{cases}$$

  where $N_{\text{moves}}$ is the number of moves.

- **Sudoku:** The score includes a base score of 2 points per non-initial filled cell, a correctness score of 10 points per correctly filled non-initial cell, and a 1000-point bonus for winning. The total score is:

$$\text{Score} = (N_{\text{filled}} \times 2) + (N_{\text{correct}} \times 10) + B_{\text{win}}, \quad B_{\text{win}} = \begin{cases} 1000 & \text{if win} \\ 0 & \text{otherwise} \end{cases}$$

  where $N_{\text{filled}}$ is the number of filled cells, and $N_{\text{correct}}$ is the number of correct fills.

- **Reversi (Othello):** The score consists of a step score (10 points per move), a piece bonus (20 points per Black piece), and an outcome bonus (1000 points for a win, 500 for a tie, 0 for a loss). The total score is:

$$\text{Score} = (N_{\text{moves}} \times 10) + (N_{\text{black}} \times 20) + B_{\text{outcome}}, \quad B_{\text{outcome}} = \begin{cases} 1000 & \text{if win} \\ 500 & \text{if tie} \\ 0 & \text{if loss} \end{cases}$$

  where $N_{\text{moves}}$ is the number of moves, and $N_{\text{black}}$ is the number of Black pieces.

- **Minesweeper:** The score comprises a step score (10 points per move), a reveal bonus (2 points per revealed non-mine cell), and a 1000-point bonus for winning. The total score is:

$$\text{Score} = (N_{\text{moves}} \times 10) + (N_{\text{revealed}} \times 2) + B_{\text{win}}, \quad B_{\text{win}} = \begin{cases} 1000 & \text{if win} \\ 0 & \text{if loss} \end{cases}$$

where $N_{\text{moves}}$ is the number of moves, and $N_{\text{revealed}}$ is the number of revealed non-mine cells.

- **Gomoku:** The score includes a base score (10 points per move by Black) and an outcome bonus (1000 points for a win, 500 for a tie, 0 for a loss). The total score is:

$$\text{Score} = (N_{\text{moves}} \times 10) + B_{\text{outcome}}, \quad B_{\text{outcome}} = \begin{cases} 1000 & \text{if win} \\ 500 & \text{if tie} \\ 0 & \text{if loss} \end{cases}$$

where $N_{\text{moves}}$ is the number of moves.

- **Chess:** The score combines a step score (10 points per White move), a piece bonus (5 times the value of captured Black pieces: Pawn=1, Knight/Bishop=3, Rook=5, Queen=9), and an outcome bonus (1000 for a win, 500 for a tie, 0 for a loss). The total score is:

$$\text{Score} = (N_{\text{moves}} \times 10) + (V_{\text{captured}} \times 5) + B_{\text{outcome}}, \quad B_{\text{outcome}} = \begin{cases} 1000 & \text{if win} \\ 500 & \text{if tie} \\ 0 & \text{if loss} \end{cases}$$

where $N_{\text{moves}}$ is the number of moves, and $V_{\text{captured}}$ is the total value of captured Black pieces.

These scoring mechanisms provide a comprehensive evaluation of the model's performance in the E2E Playing task, balancing efficiency, strategic decision-making, and successful outcomes across diverse game types.

## C  SEARCH-BASED AI OPPONENTS

In adversarial games, effective decision-making is essential as players must anticipate and counter their opponent's moves. To facilitate this, many search-based methods have established a solid foundation for developing AI opponents. Below are key approaches utilized in our framework:

- **Tic-Tac-Toe:** The AI opponent employs the **Minimax** algorithm, which recursively evaluates all possible moves to determine the optimal play. The algorithm considers both the AI's and the opponent's potential moves, aiming to maximize the AI's chances of winning while minimizing the opponent's chances. The evaluation function assesses the board state, returning positive scores for winning configurations and negative scores for losing ones. This allows the AI to select the best possible move at each turn, effectively making it a formidable opponent.

- **Sudoku:** N/A

- **Reversi:** The AI opponent in Reversi uses an **Alpha-Beta pruning** technique combined with a scoring system to evaluate board states. The AI identifies valid moves and simulates their outcomes by temporarily applying the moves to a copy of the board. The Alpha-Beta algorithm efficiently narrows down the search space, allowing the AI to maximize its score while minimizing the opponent's score. This approach enables the AI to make strategic decisions and adapt to the opponent's moves effectively.

- **Minesweeper:** N/A

- **Gomoku:** The Gomoku AI utilizes a recursive evaluation strategy to determine the best move. It evaluates the board based on potential winning patterns, assessing both its own and the opponent's configurations. The AI implements a **depth-limited search** to simulate possible moves, using an evaluation function that assigns values based on the length of contiguous pieces. This allows the AI to prioritize moves that advance its strategy while blocking the opponent's attempts to connect five in a row.

- **Chess:** The Chess AI leverages the open-source Stockfish[1] engine, which is a lightweight chess engine that implements the **Minimax** algorithm with **Alpha-Beta pruning**. This library enables the AI to evaluate board positions efficiently, considering multiple possible moves and their outcomes. The Stockfish engine uses a heuristic evaluation function to assess board states, allowing the AI to make strategic decisions and respond to the opponent's moves effectively.

---

[1]https://stockfishchess.org/

