# OpenReview forum: "Are Large Vision Language Models Good Game Players?"
_ICLR.cc/2025/Conference — ICLR 2025 Poster_

### Official Review · Reviewer_iEip · 2024-10-18

**Soundness:** 3
**Presentation:** 4
**Contribution:** 4
**Rating:** 8
**Confidence:** 3

**Summary:**

This paper presents LVLM-Playground, a game-based benchmark for evaluating Large Vision Language Models (LVLMs). Using six diverse games and four tasks (Perceiving, Question Answering, Rule Following, and End-to-End Playing), it assesses LVLMs' abilities beyond the limitations of traditional VQA benchmarks. Experiments reveal model weaknesses in handling long outputs, detailed perception, and instruction following (potentially due to RLHF), while also showing that gameplay data can improve reasoning during fine-tuning.

**Strengths:**

1. Novel Benchmark: LVLM-Playground provides a novel, game-based approach to evaluating LVLMs, offering a more dynamic and comprehensive assessment than traditional benchmarks.

2. Rigorous Methodology: The paper presents a well-structured and technically sound methodology, with detailed task formulations, evaluation metrics, and implementation of AI opponents.

3. Significant Findings:  The benchmark reveals key limitations in current LVLMs, particularly in handling complex visual details and instruction following, while also suggesting gameplay data as a potential avenue for improvement.

**Weaknesses:**

Re. Finding 3 - I find it less sound. Maybe I am missing something, but -  Yes, VLMs produce long answers, and the expected answers are short, but this is an evaluation problem, not the model’s fault. There  are several ways to overcome it - few-shot prompts with short answers, parsing the answers with regex to a closed set of answers, a model-based LLM, and so on.

**Questions:**

The LVLM-Playground framework seems highly beneficial for the community. Do the authors plan to publicly release the code and data?

Could the authors elaborate on the evaluation of the Q&A task? Are the methods I mentioned (few-shots, LLM-based evaluation, etc) relevant for overcoming the issue you mentioned?

---

> ### Author Response · Authors · 2024-11-21
> **Response to Reviewer iEip #Q.1**
>
> **Q.1** *Finding 3 - I find it less sound. Maybe I am missing something, but - Yes, VLMs produce long answers, and the expected answers are short, but this is an evaluation problem, not the model’s fault. There are several ways to overcome it - few-shot prompts with short answers, parsing the answers with regex to a closed set of answers, a model-based LLM, and so on.*
>
>
> **A.1**  We appreciate the reviewer’s insightful feedback. Regarding Finding 3, our observation highlights a difference in instruction-following capability between open-source and commercial models. Specifically, commercial models tend to produce long responses due to human preference alignment via RLHF. However, **our evaluation does NOT penalize models simply for producing lengthy answers; rather it focuses on their inability to adhere to the expected or specified response format.**
>
>   For example, in the question-answering task, we provide multiple examples as references for the expected answer format. For a question such as:
>
>   *"Which player has more pieces on the board, 'Black', or 'White'? Please respond with 'Black', 'White', or 'Equal'."*
>
>   The model is expected to choose one of these three options. However, we observed that commercial models frequently fail to follow this expected output format and instead produce verbose but irrelevant explanations that do not align with the task requirements.
>
>   Recognizing the limitations of the existing evaluation approach, we have revised the question-answering task to a **closed set of answers (multiple-choice) format** based on the reviewer’s suggestions. This modification avoids formatting issues and ensures a clearer focus on the reasoning and perception abilities that the Q&A task aims to evaluate. More details are provided in our response to the reviewer’s second question.

---

> > ### Author Response · Authors · 2024-11-21
> > **Response to Reviewer iEip #Q.2**
> >
> > **Q.2** *Could the authors elaborate on the evaluation of the Q&A task? Are the methods I mentioned (few shots, LLM-based evaluation, etc.) relevant to overcoming the issue you mentioned?*
> >
> > **A.2** We appreciate the reviewer’s constructive feedback. In light of the reviewer’s comments and suggestions from other reviewers, we recognized a limitation in the existing metric used for the question-answering task in the LVLM-Playground. Specifically, the current evaluation metric did not sufficiently decouple instruction-following capability from the perceiving and reasoning abilities, which are the primary focus of the Q&A task.
> >
> >   Since the question-answering task is designed to primarily evaluate reasoning and perception, it is inappropriate for the metric to penalize models for formatting issues arising from instruction-following errors. To address this, **we revised the question-answering generation and evaluation process, converting it from an open-ended evaluation to a closed multiple-choice format.** Models are now prompted to select one of four predefined options, thus mitigating formatting issues during evaluation.
> >
> >   ### Previous Results:
> >   | LVLMs              | TicTacToe | Reversi | Sudoku | Minesweeper | Gomoku | Chess |
> >   |--------------------|-----------|---------|--------|-------------|--------|-------|
> >   | GPT-4o            | 0.235     | 0.214   | 0.296  | 0.231       | 0.271  | 0.168 |
> >   | Gemini1.5-pro     | 0.280     | 0.277   | 0.233  | 0.196       | 0.226  | 0.232 |
> >   | Claude-3.5-sonnet | 0.216     | 0.178   | 0.224  | 0.204       | 0.245  | 0.143 |
> >   | Qwen2-vl-7b       | 0.655     | 0.439   | 0.234  | 0.273       | 0.148  | 0.314 |
> >   | Deepseek-vl-7b    | 0.304     | 0.261   | 0.130  | 0.355       | 0.211  | 0.257 |
> >   | Phi3-vl           | 0.515     | 0.314   | 0.172  | 0.262       | 0.127  | 0.232 |
> >   | LLaVA-1.6-7b      | 0.387     | 0.222   | 0.189  | 0.197       | 0.219  | 0.269 |
> >   | InternVL2-8b      | 0.675     | 0.281   | 0.297  | 0.188       | 0.127  | 0.215 |
> >
> >   Using this revised approach, we re-evaluated all models for the question-answering task. The updated results show significant improvements for most models, especially for commercial models, as they are no longer penalized for verbose or unexpected formatting. These changes address the reviewer’s suggestions regarding few-shot prompting and closed-format evaluations. Additionally, we ensured that model-based LLM evaluations (e.g., parsing outputs using regex or employing an LLM answer extractor) were integrated where appropriate, particularly for validation.
> >
> >   The updated results have been included in the revised version of our paper, with changes highlighted in blue text for clarity. We believe these updates improve the robustness of our evaluation process and address the reviewer’s concerns. Thank you again for highlighting this important point.

---

> > > ### Author Response · Authors · 2024-11-21
> > > **Response to Reviewer iEip #Q.3**
> > >
> > > **Q.3** *The LVLM-Playground framework seems highly beneficial for the community. Do the authors plan to publicly release the code and data?*
> > >
> > > **A.3** Thank you for the positive feedback and interest in our work. We are committed to open-source and **will publicly release all resources associated with this paper**, including the LVLM-Playground benchmark suite, parameter settings, prompt templates, and the results obtained by baseline models.
> > >
> > >   We hope that this work serves as a foundation for enhancing the community's understanding of the strengths and limitations of existing LVLMs, while also facilitating future research and development in this area.

---

> > > ### Comment · Reviewer_iEip · 2024-11-26
> > >
> > > Thanks for the detailed response, clarifications, and experiments. I retain my score: 8.

---

> > > > ### Author Response · Authors · 2024-11-26
> > > >
> > > > We thank the reviewer again for the effort spent during the reviewing process. We also appreciate the positive feedback and constructive suggestions, which have helped us to further improve our work.

---

> ### Author Response · Authors · 2024-11-23
> **Follow up on Discussion**
>
> Dear Reviewer iEip,
> Thank you for your valuable feedback on our submission. We hope that our responses have addressed your concerns. Please let us know if there are any remaining issues or further questions, and we will be happy to provide additional clarifications. We appreciate your time and effort during the discussion period.

---

### Official Review · Reviewer_giEs · 2024-11-03

**Soundness:** 2
**Presentation:** 3
**Contribution:** 2
**Rating:** 6
**Confidence:** 4

**Summary:**

This paper proposes a new benchmark to evaluate large vision-language models in structured environments across 4 tasks in 6 games of different difficulties. This benchmark is designed to evaluate different abilities of vision-language models including perception, reasoning, decision-making and adversary.

**Strengths:**

- This paper is well-motivated and well-written overall.
- The idea of using structured game environments to evaluate vision-language models is somewhat novel.
- Some designs of the benchmark are well thought out: for example, it includes games that test different abilities of the vision-language models and cover different difficulty levels and provides clear definitions of the complexity of each ability.
- The paper is clearly written with sufficient technical details and effective illustrations.

**Weaknesses:**

The main area of improvement for this paper is the alignment between the experimental designs and the claims. The reviewer is mainly concerned about some of the evaluation task designs and whether they actually test vision-language models’ perception or reasoning abilities like the authors claim.
- For example, the perceiving task prompts the models to output matrices that correspond to the game states, which not only tests for perception but also requires strong instruction-following ability. Similarly, while the Q&A is mostly designed to test for perception and reasoning (Figure 3), it asks the models to output answers in a particular format and therefore also requires strong instruction-following ability.
-	As the authors discussed in finding 3, all commercial models fail to match the output format or refuse to answer in the Q&A task, which means this task doesn’t actually test their perception/reasoning ability as claimed.
-	In the reviewer’s opinions, the tasks can be designed better to decouple the evaluations of perception/reasoning and instruction following abilities.
-	For example, most VLM evaluation benchmarks adopt the multi-choice Q&A format when testing for their perception/reasoning abilities. And when the answer format is free-form, VLMEvalKit -- one of the most widely used evaluation frameworks of VLMs – uses an LLM as an answer extractor, putting more emphasis on the models’ perception/reasoning abilities rather than their instruction-following ability.
-	Due to the differences in task and evaluation designs, the results reported in this paper (e.g. Table 3) regarding these models’ perception and reasoning abilities deviate a lot from existing works.
-	The authors also claim that gameplay data could enhance LVLM’s reasoning abilities but have only provided very limited evidence on a subset of one VQA benchmark.

VLMEvalKit: An Open-Source Toolkit for Evaluating Large Multi-Modality Models: https://arxiv.org/abs/2407.11691

**Questions:**

-	The reviewer appreciates that the authors conducted a human study on the difficulty rating, which mostly though not perfectly aligns with the difficulty rating defined by the authors. The reviewer wonders if the authors could also obtain human ratings for the four dimensions
-	For finding 6, did the authors use a randomly sampled subset of VQA? Does this subset test for VLMs’ reasoning ability? If not, the authors could consider using a more suitable benchmark or revising their claim.

---

> ### Author Response · Authors · 2024-11-21
> **Response to Reviewer giEs #Q.1**
>
> **Q.1** *The perceiving task prompts the models to output matrices that correspond to the game states, which not only tests for perception but also requires strong instruction-following ability.*
>
> **A.1** We thank the reviewer for the insightful comment. We agree that the perceiving task inherently tests not only perception but also instruction-following ability, as the task requires the model to output matrices corresponding to game states. However, instruction-following is a fundamental capability of LLMs, and it is linked to almost all other abilities, such as perception, reasoning, and decision-making. This interdependence makes it inherently challenging to fully isolate instruction-following from other capabilities in practical evaluations.
>
>   To address the reviewer’s concern and provide a deeper analysis, we conducted additional experiments on the perceiving task, focusing on how board density may affect model performance:
>
>   | LVLMs              | 0-45  | 46-90 | 91-135 | 136-180 | 181-225 |
>   |--------------------|-------|-------|--------|---------|---------|
>   | GPT-4o            | 0.801 | 0.553 | 0.381  | 0.320   | 0.413   |
>   | Gemini1.5-pro     | 0.864 | 0.636 | 0.464  | 0.395   | 0.412   |
>   | Claude-3.5-sonnet | 0.878 | 0.728 | 0.686  | 0.694   | 0.774   |
>   | Qwen2-vl-7b       | 0.861 | 0.543 | 0.382  | 0.264   | 0.100   |
>   | Deepseek-vl-7b    | 0.043 | 0.011 | 0.000  | 0.000   | 0.000   |
>   | Phi3-vl           | 0.210 | 0.021 | 0.019  | 0.004   | 0.005   |
>   | LLaVA-1.6-7b      | 0.073 | 0.000 | 0.000  | 0.000   | 0.000   |
>   | InternVL2-8b      | 0.891 | 0.692 | 0.421  | 0.114   | 0.097   |
>
>   Taking Gomoku as an example, we varied the number of pieces on the board across five levels, ranging from sparse to dense, and evaluated the model's perception accuracy under these conditions. Based on the results, we found that most models are highly sensitive to board density. For example, InternVL2-8B performed exceptionally well on sparse boards, even outperforming all commercial models in the 0-45 pieces set, but its performance dropped sharply to 0.097 due to unexpected outputs (such as looping behavior described in Finding 1 of the main paper) on the densest boards (181-225 pieces).
>
>   This result highlights that while the models possess instruction-following capability to output the game state in the required matrix format, **their ability to follow instructions is significantly influenced by their perception performance**. If a model struggles to accurately perceive the detailed game state, it often fails to generate the expected matrix, demonstrating that perception and instruction-following are interdependent.
>
>   Therefore, we believe the design of the perceiving task reasonably reflects and enables a fair comparison of the perception capabilities across different LVLMs, while also accounting for their instruction-following abilities in practical scenarios.

---

> > ### Author Response · Authors · 2024-11-21
> > **Response to Reviewer giEs #Q.2 Part-1**
> >
> > **Q.2** *Similarly, while the Q&A is mostly designed to test for perception and reasoning (Figure 3), it asks the models to output answers in a particular format and therefore also requires strong instruction-following ability. As the authors discussed in Finding 3, all commercial models fail to match the output format or refuse to answer in the Q&A task, which means this task doesn’t actually test their perception/reasoning ability as claimed. In the reviewer’s opinion, the tasks can be designed better to decouple the evaluations of perception/reasoning and instruction-following abilities. For example, most VLM evaluation benchmarks adopt the multi-choice Q&A format when testing for their perception/reasoning abilities. When the answer format is free-form, VLMEvalKit -- one of the most widely used evaluation frameworks of VLMs – uses an LLM as an answer extractor, putting more emphasis on the models’ perception/reasoning abilities rather than their instruction-following ability.*
> >
> > **A.2-1** We appreciate the reviewer’s thoughtful comments. Our Q&A task already follows a structure similar to a multiple-choice format, though it does not explicitly label options as A, B, C, or D. For example, as listed in Appendix B.2, many questions explicitly provide the valid candidate answers in the prompt. Examples include:
> >
> >   - What is the symbol in row A, column 3? Please respond with **'X', 'O', or 'empty'**.
> >   - What is the number in row 2, column 5? Please respond with **a single Arabic numeral or 'empty'**.
> >   - Which player has more pieces on the board, 'Black' or 'White'? Please respond with **'Black', 'White', or 'equal'**.
> >
> >   This design simplifies the task by clearly outlining the valid responses, significantly reducing the difficulty of adhering to the required format. This may explain why open-source models perform reasonably well on the task. Therefore, we believe the question format itself may not be the primary reason for the commercial models' challenges in matching the expected output format.
> >
> > However, we agree with the reviewer that it is essential to further decouple perception and reasoning capabilities from instruction-following ability to eliminate confounding factors. In response to the reviewer’s suggestion, **we have modified all Q&A tasks to a strict multiple-choice format**, where each question explicitly includes no more than four candidate options. This format minimizes reliance on instruction-following while keeping the focus on perception and reasoning. We conducted new experiments under this revised setting, and the updated results are as follows:
> >
> > | LVLMs              | TicTacToe | Reversi | Sudoku | Minesweeper | Gomoku | Chess |
> >   |--------------------|-----------|---------|--------|-------------|--------|-------|
> >   | GPT-4o            | 0.235     | 0.214   | 0.296  | 0.231       | 0.271  | 0.168 |
> >   | Gemini1.5-pro     | 0.280     | 0.277   | 0.233  | 0.196       | 0.226  | 0.232 |
> >   | Claude-3.5-sonnet | 0.216     | 0.178   | 0.224  | 0.204       | 0.245  | 0.143 |
> >   | Qwen2-vl-7b       | 0.655     | 0.439   | 0.234  | 0.273       | 0.148  | 0.314 |
> >   | Deepseek-vl-7b    | 0.304     | 0.261   | 0.130  | 0.355       | 0.211  | 0.257 |
> >   | Phi3-vl           | 0.515     | 0.314   | 0.172  | 0.262       | 0.127  | 0.232 |
> >   | LLaVA-1.6-7b      | 0.387     | 0.222   | 0.189  | 0.197       | 0.219  | 0.269 |
> >   | InternVL2-8b      | 0.675     | 0.281   | 0.297  | 0.188       | 0.127  | 0.215 |
> >
> > From the updated results, we observed that open-source models still maintain their performance, with some achieving even higher accuracy due to the simplified multiple-choice format. However, while commercial models such as GPT-4o no longer exhibit near-zero accuracy, their performance remains close to random guessing (around 25% for four-choice questions). This suggests that the free-form nature of the previous question format might not be the main cause of their poor performance in this task. As such, the conclusions drawn in Finding 3 of the main paper still hold.

---

> > > ### Author Response · Authors · 2024-11-21
> > > **Response to Reviewer giEs #Q.2 Part-2**
> > >
> > > **A.2-2** To further analyze the errors, we categorized them into two main types:
> > >
> > >   1. **Incorrect choice:** The model selects an incorrect answer from the candidate pool.
> > >   2. **Unexpected format:** The model's response deviates entirely from the candidate options or the required format (e.g., verbose or nonsensical outputs).
> > >
> > >   | LVLMs              | Incorrect Selection | Unexpected Format |
> > >   |--------------------|---------------------|-------------------|
> > >   | GPT-4o            | 4541                | 44                |
> > >   | Gemini1.5-pro     | 4517                | 39                |
> > >   | Claude-3.5-sonnet | 4733                | 57                |
> > >   | Qwen2-vl-7b       | 3937                | 0                 |
> > >   | Deepseek-vl-7b    | 4479                | 3                 |
> > >   | Phi3-vl           | 4377                | 1                 |
> > >   | LLaVA-1.6-7b      | 4515                | 2                 |
> > >   | InternVL2-8b      | 4217                | 0                 |
> > >
> > >   Based on the above results, we found that by converting the Q&A task into an explicitly multiple-choice format, **errors caused by unexpected output formats are **no longer the primary factor contributing to the low performance of the models**, though they still have some impa, especially for commercial models. Therefore, we believe that the revised Q&A task setting is reasonable for evaluating perception and reasoning abilities. It has, to a certain extent, been decoupled from instruction-following requirements, where unexpected output formats are less likely to be penalized.

---

> > > > ### Author Response · Authors · 2024-11-21
> > > > **Response to Reviewer giEs #Q.3-Q.5**
> > > >
> > > > **Q.3** *Due to the differences in task and evaluation designs, the results reported in this paper (e.g. Table 3) regarding these models’ perception and reasoning abilities deviate a lot from existing works.*
> > > >
> > > >  **A.3** We appreciate the reviewer for raising this important point. We acknowledge that some findings reported in this paper, such as those in Table 3, may not fully align with results from other existing evaluation benchmarks, particularly for the Q&A task. However, this deviation highlights one of our key motivations: *While LVLMs have demonstrated exceptional performance on many existing benchmarks, especially those focusing on daily, natural scenarios, their generalization capabilities to less-representative scenarios remain unclear.*
> > > >
> > > >   LVLMs are extensively trained on datasets that predominantly cover common, naturalistic contexts. As a result, evaluating them in these scenarios may not fully reflect their true capabilities. To address this gap, we designed a game-based evaluation framework that introduces structured, less-representative scenarios—settings that are less likely to have been encountered during training. **This allows us to probe the models' generalization abilities and uncover limitations that might not surface in traditional benchmarks**. We believe that this evaluation framework offers new insights into the strengths and weaknesses of existing LVLMs.
> > > >
> > > > **Q.4** *The reviewer appreciates that the authors conducted a human study on the difficulty rating, which mostly though not perfectly aligns with the difficulty rating defined by the authors. The reviewer wonders if the authors could also obtain human ratings for the four dimensions.*
> > > >
> > > > **A.4**  We appreciate the reviewer’s thoughtful suggestion. To further validate the evaluation settings, particularly the difficulty levels of each game across the four dimensions (perception, reasoning, decision-making, and adversary), we conducted an additional human study involving 10 volunteers.
> > > >
> > > >   Volunteers were asked to perform tasks themselves to understand the difficulty of perceiving, question-answering, and rule-following tasks. Additionally, to assess adversary skills, volunteers participated in competitive matches against the search-based AI opponent. To ensure consistency, we provided detailed game rules and guidelines beforehand. The aggregated feedback is summarized below:
> > > >
> > > >   |               | Tic-Tac-Toe | Reversi | Sudoku | Minesweeper | Gomoku | Chess |
> > > >   |---------------|-------------|---------|--------|-------------|--------|-------|
> > > >   | Perception    | 1.00        | 2.20    | 4.60   | 4.50        | 4.80   | 3.90  |
> > > >   | Reasoning     | 1.00        | 2.10    | 3.90   | 4.70        | 4.50   | 4.80  |
> > > >   | Decision      | 1.20        | 2.50    | 2.30   | 4.70        | 4.90   | 5.40  |
> > > >   | Adversary     | 1.00        | 2.50    | N/A    | N/A         | 2.60   | 3.90  |
> > > >
> > > >   The results demonstrate **trends consistent with our predefined difficulty ratings**, validating the robustness of our framework. We have incorporated these updates in the revised paper with detailed descriptions and charts summarizing the results.
> > > >
> > > > **Q.5** *The authors also claim that gameplay data could enhance LVLM’s reasoning abilities but have only provided very limited evidence on a subset of one VQA benchmark. For finding 6, did the authors use a randomly sampled subset of VQA? Does this subset test for VLMs’ reasoning ability? If not, the authors could consider using a more suitable benchmark or revising their claim.*
> > > >
> > > > **A.5**  We thank the reviewer for the thoughtful comment. Inspired by findings that specific types of instruction data, such as coding data, can enhance the reasoning capabilities of LVLMs, we conducted a preliminary exploration to investigate whether game-based instruction data could similarly improve generic reasoning abilities in LVLMs. For this, we **randomly** sampled a subset from the VQA dataset and performed a preliminary test using MiniGPT-4, chosen for its simplicity and ease of implementation.
> > > >
> > > >   However, after considering the reviewer’s feedback, we acknowledge that the current results, based solely on the MiniGPT-4 model, are insufficient to draw definitive conclusions. As such, we have removed this finding (Finding 6) from the revised version of the paper and will leave this topic for future exploration.
> > > >
> > > >   We believe this adjustment does not impact the overall contribution of LVLM-Playground. The benchmark's core contributions lie in its comprehensive evaluation framework for perception, reasoning, decision-making, and adversary abilities across structured game environments. These contributions remain robust and provide insights for advancing the evaluation and development of LVLMs.

---

> ### Author Response · Authors · 2024-11-23
> **Follow up on Discussion**
>
> Dear Reviewer giEs,
> Thank you for your valuable feedback on our submission. We hope that our responses have addressed your concerns. Please let us know if there are any remaining issues or further questions, and we will be happy to provide additional clarifications. We appreciate your time and effort during the discussion period.

---

> ### Author Response · Authors · 2024-11-25
>
> Dear Reviewer giEs,
>
> As the author-reviewer discussion period is nearing its end, we were wondering if our response has adequately addressed your concerns. We are open to providing any further clarifications if needed and greatly value your feedback.
>
> Thank you again for the effort you have dedicated to reviewing our paper and for offering many constructive suggestions that have helped us further improve our work.

---

> > ### Author Response · Authors · 2024-11-26
> >
> > Dear Reviewer giEs,
> >
> > As the discussion period is nearing its end, we wanted to kindly follow up to see if our responses have addressed your concerns. It is the last day for authors to revise the paper based on reviewers’ feedback, and we would greatly appreciate any further suggestions or clarifications you might have.
> >
> > Thank you again for your time and valuable input throughout the review process.

---

> ### Comment · Reviewer_giEs · 2024-11-26
>
> Thank the authors for the detailed responses and updated results based on the suggestions! They addressed some of my concerns, with a few ones left:
>
> Regarding Q.2 Part-1, I'm still skeptical about the numbers for commercial models -- are they low because the answer extraction can be improved? or do the authors believe that the results suggest commercial models have worse perception abilities than some of the open source models?
>
> The reviewer also appreciates the error analysis in A.2-2 but still has questions: was the error analysis done manually or automatically (and if the latter, how)? Also, "Incorrect selection" is a very broad category and lisiting out the error count doesn't help understand why commercial models have higher error rates than open source ones, making it still unclear to the reviewer if their worse performance is due to weaker perception, worse generalizability or the task formulation -- can the authors provide deeper qualitative analysis into these particular results?

---

> > ### Author Response · Authors · 2024-11-26
> > **Response to Reviewer giEs's Follow-up Question Part-1**
> >
> > We thank the reviewer for the thoughtful comments and questions.
> >
> > **Regarding Error Types.**
> >
> > The error analysis was conducted automatically using a regex-based parser to extract the answer from the LVLMs' responses. Typically, the model's response follows a format like:
> >
> > *“Your Option: A”*
> >
> > In this case, the parser extracts the answer **'A'**. However, there are instances where LVLMs fail to produce a response in the expected format. Instead, they might refuse to answer (e.g., *"Sorry, but I cannot assist with that"*) or generate outputs that cannot be parsed. These cases are categorized under **Unexpected Format** errors.
> >
> > The reasons for categorizing errors only into **Incorrect Selection** and **Unexpected Format** are as follows:
> >
> > - By converting the Q&A task into a multiple-choice format, LVLMs are now expected to output only one of the predefined options (e.g., A, B, C, or D). While this design simplifies the task and decouples strict instruction-following from reasoning and perception abilities, it inherently focuses the evaluation on the correctness of the selected option. As a result, the reasoning behind the choice of a specific option is not explicitly evident from the response alone, and we therefore group all such errors under the broader category of **Incorrect Selection**.
> > - The original Q&A task was semi-open-ended, but it was converted into an explicit multiple-choice format to investigate whether formatting issues were a major cause of low accuracy. While commercial models are more prone to producing unexpected format responses (mainly due to occasionally refusing to answer), our analysis shows that the primary reason for their lower accuracy lies in **Incorrect Selection**, rather than formatting errors.

---

> ### Author Response · Authors · 2024-11-26
> **Response to Reviewer giEs's Follow-up Question Part-2.1**
>
> **Regarding Deeper Qualitative Analysis**
>
> We acknowledge that **Incorrect Selection** is a broad category that may not fully capture the weaknesses in perception, generalizability, or task formulation. To address the reviewer's concern and provide deeper qualitative insights, we conducted an additional analysis where models were prompted to explain their reasoning after making a selection. This approach helps us better understand why specific options were chosen.
>
> We used the Tic-Tac-Toe game as an example for this analysis. Specifically, we modified the task prompt format from:
>
> ```
> ....
> Your Option:
> ```
>
> to
>
> ```
> .... Then explain the reason why you chose that option.
> Your Option:
> Explanation:
> ```
>
> We added an **additional instruction**: *"Then explain the reason why you chose that option."* to the end of the original prompt.
>
> This modification **explicitly requests the model to provide a reasoning process** alongside its answer, enabling a more detailed qualitative analysis of the model's decision-making. Since this explanation-based prompt sometimes broke the multiple-choice structure and the original regex-based parser failed to extract correct answers, we conducted a manual review of the responses and primarily focused on qualitative analysis.
>
> By enforcing the models to provide an explanation for their choice, several limitations in their reasoning and instruction-following abilities become apparent, aligning with our initial findings in the Q&A task. From the new qualitative results, we found that commercial models often struggle to fulfill the given instructions and are prone to providing nonsensical or irrelevant responses. For example:
>
> - **Claude:** The model repeatedly provides responses such as, *"I apologize, but there is no multiple-choice question provided in your message for me to answer."* or *"Given this game state, if I were to suggest a move for the 'O' player, I would recommend: B2."* These examples show that the model fails to understand and follow the given instructions.
> - **Gemini:** Similar issues arise, where the model anticipates playing the game instead of answering the question. It often suggests moves such as, *"Placing an X in A1 will complete three Xs in a row (A1, B1, C1) making X the winner,"* or *"Placing an X in A1 will create a three-in-a-row victory for X. This is the only available move that achieves an immediate win."*
> - **GPT-4o:** The model also tends to interpret the task as gameplay instead of answering the questions. For example, it produces responses like, *"To determine the best move, the aim is to either win the game or block the opponent from creating a winning line. As you are playing as 'O,' you should look for opportunities to create a line of three O's or prevent 'X' from doing so."* This demonstrates a clear lack of adherence to the instructions provided in the prompt.
>
> On the other hand, when we ran the **open-source LVLMs** with **the exact same prompt**, they were more likely to follow the instructions and answer the questions correctly. For example:
>
> - **Qwen2-VL:** The model consistently generates reasonable explanations for its choices, even when selecting incorrect answers. For example:
>   *"The red marks are represented by the letter 'X'. In row C, there are two red marks, one in cell C1 and another in cell C3. Therefore, the correct answer is B. 2."* and
>   *"There are three 'O's on the board: one in row A, column 1; one in row C, column 1; and one in row C, column 3."*
>
> - **InternVL2:** Similarly, this model demonstrates a good understanding of the task and adheres to the Question Answering baseline, providing logical reasoning. For example:
>   *"Upon examining the Tic-Tac-Toe board, we can see that there are three 'O's present. One 'O' is in the top right corner (position 2), another 'O' is in the middle left corner (position 2), and the third 'O' is in the bottom right corner (position 2). Therefore, the correct answer is B, as there are three 'O's on the board."*
>
> Although these open-source models make **incorrect choices** due to poorer perception capability, they generally **follow the instructions and explain their reasoning in a coherent manner**.

---

> ### Author Response · Authors · 2024-11-26
> **Response to Reviewer giEs's Follow-up Question Part-2.2**
>
> Based on the above qualitative analysis, we draw the following observations to address the reviewer's concerns:
>
> - Since the commercial models generally performed much better on the perception task than the open-source models, we do **not** believe the performance gap in the Q&A task is due to weaker perception abilities. Instead, it appears to stem from **difficulties in understanding and following the question-answering instructions**.
>
> - The multiple-choice format forces models to choose from predefined options, requiring them to follow the instructions. Although this enforcement ensures that commercial models select an answer from the candidate options, it may **obscure the fact that they do not authentically understand or adhere to the instructions**. This often results in responses that resemble random selections from the available options, which explains their low performance.
>
> - We believe the commercial models' low performance may stem from the gap in task prompts. Many studies have shown that discrepancies between training and testing prompts can lead to poorer performance for LVLMs. As mentioned in the introduction section of our paper, existing evaluation frameworks often rely on varying prompts, which can significantly impact model performance. This motivates LVLM-Playground to provide a more consistent evaluation environment, where all models are evaluated using the **exact same task prompts**. In LVLM-Playground, we intentionally use casual, consistent task descriptions rather than meticulously designed, model-specific prompts. **We believe that with more carefully tailored prompts, commercial models might achieve significantly better performance.** However, the existing task prompts may favor open-source models, which are less influenced by RLHF, while highlighting **potential weaknesses** in commercial models to generalize across different prompt styles.
>
> Since the response system does not support figure attachments and has character limitations, **we have included these new qualitative results and findings in the revised paper, as well as a few example evaluation logs of each model**. The reviewer may refer to the appendix, starting from **Page 43 to 48**, for the complete analysis and examples.
>
> We sincerely appreciate the reviewer's constructive comments and suggestions. We will extend these deeper qualitative analysis to all other games in the LVLM-Playground and provide all evaluation logs in the final release of LVLM-Playground.

---

> > ### Comment · Reviewer_giEs · 2024-11-27
> >
> > The reviewer sincerely appreciates the authors for their detailed responses and diligence in the rebuttal process! However, the reviewer is still unconvinced about these results and analyses.
> >
> > The authors mentioned that their analysis suggests commercial models are worse than open source models because (1) their failure in understanding and following the QA instructions; and (2) they do NOT generalize well to the specific task prompts used, which are favorable to open source models -- both points are very specific to the particular task formulation in this work instead of generalizable findings about commercial vs. open source models (e.g. it doesn't mean commercial models are always worse than open source models in following QA in general) and might be misleading if such general claims are made in the paper.
> >
> > Further, the authors even mentioned the commercial models "tend to interpret the task as gameplay" instead of question answering -- doesn't it mean that commercial models could be better game players (to answer the question posed in the paper title)? but these results could mislead readers to believe they are not. It makes the reviewer doubts if the task formulation in the paper truly represent gameplay.
> >
> > The reviewer also believes the answer extraction and error analysis can be improved -- it'd be great to conduct even a small-scale manual evaluation with a subset of examples. However, due to limited time in rebuttal, the reviewer understands that it's unreasonable to ask for more analysis.
> >
> > Therefore, the reviewer would like to maintain their score even though they'd be open to increase it if the above questions were further addressed thoughtfully.

---

> ### Author Response · Authors · 2024-11-27
> **Response to Reviewer giEs - Part 1**
>
> We sincerely thank the reviewer for their thoughtful feedback and valuable insights. We understand the concerns raised and would like to address them as follows:
>
> **Q.1** *The authors mentioned that their analysis suggests commercial models are worse than open source models because (1) their failure in understanding and following the QA instructions; and (2) they do NOT generalize well to the specific task prompts used, which are favorable to open source models -- both points are very specific to the particular task formulation in this work instead of generalizable findings about commercial vs. open source models (e.g. it doesn't mean commercial models are always worse than open source models in following QA in general) and might be misleading if such general claims are made in the paper.*
>
> **A.1** We appreciate the reviewer’s thoughtful comments. First, **the authors would like to clarify that we never made any general claim that commercial models are always worse than open-source models in our paper**. In fact, **our experimental results show that commercial models generally outperform open-source models across most tasks**, except for some Q\&A tasks. Based on this observation, we hypothesized that RLHF might hinder instruction-following ability in specific cases based on detailed qualitative analysis, but we did not make any strong conclusion or general claim that commercial models are universally inferior to open-source models under such scenarios.
>
> When we first observed the worse performance of commercial models on the Q\&A task, we were similarly surprised and conducted **multiple validations to ensure the reliability of the results**. Additionally, following the reviewer’s suggestion, we converted the semi-open-ended Q\&A task into a multiple-choice format to avoid potential penalties due to template-following issues. However, under the existing settings, all verified experimental results consistently demonstrated that commercial models show a performance gap compared to open-source models in this specific Q\&A task.
>
> We acknowledge the general expectation that commercial models typically outperform open-source models in most tasks. However, as the reviewer rightly noted, “it doesn't mean commercial models are always worse than open-source models,” it could also mean that “**open-source models may not always be worse than commercial models in certain specific contexts.**” We appreciate this nuance and understand the reviewer’s concern about whether our findings, based on specific task prompts and formulations, are generalizable. We address the reviewer's concerns as follows:
>
> - **We set the task prompts at the very beginning of this study**, and once the prompts for each game were established, **we intentionally did not modify them or introduce prompt engineering to optimize the performance of specific models**. Our goal was to **obtain vanilla results that minimize human intervention** and reflect the models' inherent capabilities. As we emphasized in the introduction, many LVLMs tune their system prompts to optimize performance for specific tasks during evaluation. While we could similarly enhance the performance of commercial models by fine-tuning systems and task prompts, we believe that such modifications could lead to misleading results, suggesting strong performance in specific tasks while lacking true generalizability.
> - All our analyses are based on the specific experimental settings provided by the LVLM-Playground framework, particularly the structured game environments and tasks designed within it. We do not aim to make a general claim about universal model performance; instead, as clarified in our motivation, **our goal is to explore the challenges current LVLMs encounter under structured game-based environments**. Based on the results and findings, **we do not believe the results in the paper are misleading but rather reflect the models' behavior in this specific context**.
>
> In summary, we acknowledge that the failures of commercial models in a single QA task within LVLM-Playground may not reflect a universal performance gap between commercial and open-source models. This is why we divided the evaluation into different sub-tasks, such as perception, rule-following, and end-to-end gameplay, where **commercial models generally outperform open-source models as expected**. For the Q\&A task, we believe the current experiments highlight **some potential limitations in existing LVLMs**, which are not confined to commercial models but also apply to open-source ones. Therefore, we retained the original results based on the consistent, initial task prompts rather than optimizing performance through meticulous prompt engineering. **We do not consider this approach misleading but rather a reflection of the models' behavior under a uniform and controlled evaluation framework.**

---

> ### Author Response · Authors · 2024-11-27
> **Response to Reviewer giEs - Part 2**
>
> **Q.2** *Further, the authors even mentioned the commercial models "tend to interpret the task as gameplay" instead of question answering -- doesn't it mean that commercial models could be better game players (to answer the question posed in the paper title)? but these results could mislead readers to believe they are not. It makes the reviewer doubts if the task formulation in the paper truly represent gameplay.*
>
> **A.2** It is true that commercial models tend to interpret the task as gameplay instead of question-answering. However, such responses do **NOT** demonstrate that commercial models are already proficient game players.
>
> - The Q&A task was not designed to evaluate the general gameplay capability of the models but rather their ability to understand game states and game rules. We believe this is a critical component of gameplay. **Therefore, providing gameplay-related responses to these tasks does not indicate that the models can play the games effectively.**
> - To assess end-to-end game-playing capabilities, we designed a specific E2E Playing task for this purpose. In this task, all existing LVLMs, whether commercial or open-source, **failed to outperform a basic search-based AI opponent** even in the simplest game, Tic-Tac-Toe. This **highlights their limitations in handling multi-modal input environments based on game screenshots**. It also reflects the misalignment between their multi-modal reasoning capabilities and their pure text-based reasoning capabilities.
> - In addition, as per our response to **Reviewer ciTF's #Q.3 regarding the intermediate metric for E2E gameplay**, the results demonstrate that **commercial models outperform all open-source models under the intermediate metric**. Although they struggle to complete full gameplay for games like Sudoku and Gomoku, this experiment suggests that commercial models exhibit stronger capabilities in gameplay-related tasks compared to open-source models under the E2E setting. We have incorporated these additional results into the paper, ensuring clarity. Therefore, **we do not believe our paper or experiments in a way mislead readers into thinking that open-source models are better suited for game environments than commercial models.**
>
> Given the performance across these subtasks, which separately evaluate different aspects of gameplay, **the existing models performed poorly in end-to-end gameplay**. Therefore, **the authors do not believe the results or the task formulations are misleading**. Instead, these findings provide an accurate reflection of the models' limitations in structured, multi-modal environments.

---

> > ### Author Response · Authors · 2024-11-27
> >
> > Finally, we sincerely appreciate the reviewer for their thoughtful and dedicated efforts during the review process, which have greatly helped us understand the strengths and areas for improvement in our paper! The reviewer's feedback has given us the opportunity to refine our work and enhance its clarity and rigor. We will revise some claims, particularly those related to the comparison between open-source and commercial models, to make them clearer and avoid any potential confusion that could mislead readers.

---

> > > ### Comment · Reviewer_giEs · 2024-11-28
> > >
> > > The reviewer thanks the authors for their diligence in the responses. The reviewer thinks that the submission would meet the acceptance bar if the authors revised their claims and included more nuanced discussion on the results of commericial vs. open-source models. And the reviewer believes that the authors will do so based on their responses during the rebuttal. Therefore, the reviewer has increased their score to 6.

---

> > > > ### Author Response · Authors · 2024-11-28
> > > >
> > > > We sincerely appreciate the reviewer's continued effort and thoughtful, constructive feedback throughout the review process. These insights have greatly contributed to improving the quality and clarity of our work. We are especially grateful for the reviewer’s recognition of the strengths and contributions of our paper. As suggested, we will carefully refine our claims and provide a more nuanced discussion of the results comparing commercial and open-source models to further enhance the manuscript. Thank you once again for your valuable input and support.

---

### Official Review · Reviewer_2f5P · 2024-11-03

**Soundness:** 3
**Presentation:** 3
**Contribution:** 3
**Rating:** 6
**Confidence:** 4

**Summary:**

The primary focus of this paper is the evaluation of “LVLMs” (Large Vision Language Models).  The authors motivate the work by noting that prior benchmarks have three key shortcomings: (i) they fail to assess detailed visual reasoning, (ii) they suffer from data contamination and (iii) they fail to assess multi-turn reasoning.  In response, the authors introduce an evaluation framework called LVLM-Playground that uses 6 games (TicTacToe, Reversi, Sudoku, Minesweeper, Gomoku and Chess) to measure LVLM capabilities. Using these games, the authors assess model performance along 4 axes: perception (i.e., accurate inference of the board state), rule following (whether a given model takes legal moves), question answering (whether the model can answer questions about a game state), and gameplay.  Through their experiments, the authors demonstrate that while frontier models are best at perception, they are (somewhat surprisingly) weaker than open-source models on question-answering tasks. This latter result is investigated and attributed to the influence of post-training (via RLHF). All models completely fail to achieve reasonable gameplay performance across the game suite against strong opponents.

**Strengths:**

Originality: Leaving aside concurrent work (mentioned below), this is the first work to extend simple game-based evaluations for LLMs to include a detailed evaluation of perception capabilities.

Significance: By introducing a visual component into simple game-based evaluation, the authors open up an interesting avenue of possibilities for efficient evaluation of model capabilities. Board games (such as chess in particular) are well-suited for benchmark generation since simulators provide automatic ground-truth and their large state space renders them relatively immune to contamination. The fact that frontier models struggle on the Chess perception task (Table 2) indicates that there is scope to guide future model development with this approach. As such, I think this work will provide a useful point of reference for the community.

Clarity: By highlighting challenges with previous benchmarks, the problem is clearly motivated. More broadly, the paper is clearly well-written, and the figures do a good job of communicating key information. Figure 4, for example, communicates the key takeaways of the paper in a compact way.

Quality: Beyond simply using games to construct a benchmark, I think the construction of several game-related subtasks is particularly useful. In particular, examining model capabilities on each game with respect to perception, Q&A and rule following is a nice way to extract additional information beyond end-to-end game playing ability.

**Weaknesses:**

1. One concern relates to the scale of the overall contribution. As the authors note, there are previous efforts such as SmartPlay (Wu et al., 2023) and GAMA-Bench (Huang et al., 2024) that explore the use of games for evaluating LLMs. The SmartPlay work, for example, also proposes a suite of games and assesses the individual capabilities (such as instruction following, reasoning and planning). As the authors highlight, it is true that these works only operate in text-space (SmartPlay converts image input into text-based visual descriptions for games that require it, rather than feeding in RGB images to the models). This is an important distinction, particularly given the focus of the current submission on assessing detailed perception. Nevertheless, the proximity of prior work somewhat diminishes the contribution of this submission.

2. A second concern relates to the breadth and depth of the analysis. On the one hand, I think the authors did an excellent job of conducting qualitative analysis (both in the form of the findings described in Section 4.1 and the observations contained within Section D of the Appendix).  On the other hand, I think the paper could have been improved by conducting more extensive quantitative analysis to provide further insight. To give a couple of concrete examples: (i) It would be interesting to see how perception performance on games like Gomoku and Chess varies as a function of board complexity (number of pieces on the board); (ii) It would be insightful to perform a manual categorization of a sample of errors on the question-answering task to inform the reader about the distribution of error types that are harming model performance.

**Questions:**

1. I’m curious to better understand the motivation behind the authors’ choice of using "unbeaten rate" to measure gameplay performance? Given access to games simulators which enable repeated games, this seems like a very high variance estimator of capability. Moreover, it is tightly anchored to the choice of opponent implementation for each game.  Did the authors consider alternatives (for example, creating a pool of opponents at different thresholds and computing Elo ratings?)

2. Related to the first question, did the authors consider evaluation against non-search based opponents? Taking Tic-Tac-Toe as an example, presumably minimax will produce optimal play, so the best that can be achieved is a sequence of draws?

Suggestion: I don’t list this point as  a “weakness” since it is concurrent work, but I would suggest citing relevant pre-prints from the first half of 2024 exploring similar directions. Examples include (1) GTBench (Duan et al., 2024), which also concludes that LLMs struggle against search-based strategies (e.g. MCTS) in games like Tic-Tac-Toe. For this reason, they also include a “random agent” baseline to illustrate that the models have achieved a basic level of competency.  (2) GameBench (Costarelli et al., 2024), which evaluates LLMs across 9 game environments and uses visual input (for models that support this).

References:
- J. Duan et al., "GTBench: Uncovering the strategic reasoning limitations of LLMs via game-theoretic evaluations." arXiv preprint arXiv:2402.12348 (2024)
- Costarelli, Anthony, et al. "GameBench: Evaluating Strategic Reasoning Abilities of LLM Agents." arXiv preprint arXiv:2406.06613 (2024).

Minor typos/suggestions:
- Minor typo on L379 (missing parenthesis)
- Claude “soonet” is presumably intended to be sonnet in table 4.1 (this typo appears in multiple various places in the paper)
- This is a minor point, but I would suggest citing MuZero in the related work section. This is perhaps the canonical example of achieving very strong performance across multiple games (including Chess, which is used in the current submission) while also being capable of processing RGB input on Atari games by leveraging both learning and search.  Reference: Schrittwieser et al., “Mastering atari, go, chess and shogi by planning with a learned model” (Nature, 2020)

---

> ### Author Response · Authors · 2024-11-21
> **Response to Reviewer 2f5P #Q.1**
>
> **Q.1** *One concern relates to the scale of the overall contribution. As the authors note, there are previous efforts such as SmartPlay (Wu et al., 2023) and GAMA-Bench (Huang et al., 2024) that explore the use of games for evaluating LLMs. The SmartPlay work, for example, also proposes a suite of games and assesses the individual capabilities (such as instruction following, reasoning, and planning). As the authors highlight, it is true that these works only operate in text space (SmartPlay converts image input into text-based visual descriptions for games that require it, rather than feeding in RGB images to the models). This is an important distinction, particularly given the focus of the current submission on assessing detailed perception. Nevertheless, the proximity of prior work somewhat diminishes the contribution of this submission.*
>
>  **A.1** We appreciate the reviewer's thoughtful feedback and acknowledgment of the importance of evaluating LLMs in game-based environments. While it is true that previous works, such as SmartPlay (Wu et al., 2023) and the concurrent GAMA-Bench (Huang et al., 2024), have explored the use of games for evaluating LLMs, our work introduces significant distinctions and contributions that go beyond these efforts. Specifically, the major differences and advancements provided by our submission include:
>
>   - **Multi-modal Input.** Existing works, such as SmartPlay and GAMA-Bench, focus exclusively on pure-text language models and rely on text-based descriptions of game states as inputs. However, this design limits its applicability in multi-modal scenarios and overlooks many unique challenges that arise from processing visual inputs. To bridge this gap, our framework incorporates multi-modal input by directly using screenshots of game states instead of converting them into text. This approach enables the evaluation of detailed perception and visual reasoning capabilities of LVLMs in a way that purely text-based systems cannot.
>
>   - **Incorporation of Competitive Environment.** The games selected in existing works are predominantly single-player, focusing only on individual capabilities without considering the competitive abilities of LLMs. In contrast, 4 out of the 6 games in our framework are two-player competitive games. These games require LVLMs to operate in adversarial settings, where they must not only strategize effectively but also counter the opponent's moves. This focus on competition highlights a novel and critical skill set not addressed in prior works.
>
>   - **Granular Assessment.** Unlike SmartPlay and GAMA-Bench, which primarily assess end-to-end game-playing performance, our work decomposes the evaluation into multiple sub-tasks. These include *perception*, *question answering*, *rule-following*, and *end-to-end gameplay*. This decomposition enables a more granular assessment of different LVLM capabilities in game environments. While SmartPlay and GAMA-Bench provide only overall game performance metrics, our approach allows for a deeper understanding of LVLMs' strengths and weaknesses across distinct abilities.
>
>   - **Systematic Definition of Evaluation Framework.** We systematically define the abilities required for game-based tasks, including *perception*, *reasoning*, *decision-making*, and *adversarial gameplay*, along with their level of difficulty in each game. This comprehensive framework enables multi-faceted evaluation and facilitates in-depth analysis, helping identify specific limitations of existing LVLMs. Such insights are crucial for guiding the future development and enhancement of LVLMs.
>
>   Therefore, although our work shares some similarities with existing works on game-based environment evaluation for LLMs, the fact that we go beyond merely adapting previous text-based approaches to a multi-modal setting highlights the novelty of our contributions. Our work not only introduces a multi-modal evaluation framework but also provides new insights and unveils unique challenges, offering fresh perspectives on LVLM evaluation and development. We have incorporated these distinctions into the related work sections of the updated version of our paper.

---

> > ### Author Response · Authors · 2024-11-21
> > **Response to Reviewer 2f5P #Q.2 - (i)**
> >
> > **Q.2** *A second concern relates to the breadth and depth of the analysis. On the one hand, I think the authors did an excellent job of conducting qualitative analysis (both in the form of the findings described in Section 4.1 and the observations contained within Section D of the Appendix). On the other hand, I think the paper could have been improved by conducting more extensive quantitative analysis to provide further insight. To give a couple of concrete examples: (i) It would be interesting to see how perception performance on games like Gomoku and Chess varies as a function of board complexity (number of pieces on the board); (ii) It would be insightful to perform a manual categorization of a sample of errors on the question-answering task to inform the reader about the distribution of error types that are harming model performance.*
> >
> > **A.2 (i)** We thank the reviewer for recognizing our qualitative analysis efforts and appreciate the valuable suggestions for further quantitative analysis. In response, we conducted additional experiments to address the points raised:
> >
> > **(i) Perception Performance vs. Board Complexity:**
> >
> > To evaluate the model's perception capability under varying board complexity, we analyzed performance as a function of the number of pieces on the board:
> >
> >   - For **Gomoku**, we varied the number of pieces (0–225 pieces) on the board and recorded the perception accuracy, as shown in the following table:
> >
> >     | LVLMs              | 0-45  | 46-90 | 91-135 | 136-180 | 181-225 |
> >     |--------------------|-------|-------|--------|---------|---------|
> >     | GPT-4o            | 0.801 | 0.553 | 0.381  | 0.320   | 0.413   |
> >     | Gemini1.5-pro     | 0.864 | 0.636 | 0.464  | 0.395   | 0.412   |
> >     | Claude-3.5-sonnet | 0.878 | 0.728 | 0.686  | 0.694   | 0.774   |
> >     | Qwen2-vl-7b       | 0.861 | 0.543 | 0.382  | 0.264   | 0.100   |
> >     | Deepseek-vl-7b    | 0.043 | 0.011 | 0.000  | 0.000   | 0.000   |
> >     | Phi3-vl           | 0.210 | 0.021 | 0.019  | 0.004   | 0.005   |
> >     | LLaVA-1.6-7b      | 0.073 | 0.000 | 0.000  | 0.000   | 0.000   |
> >     | InternVL2-8b      | 0.891 | 0.692 | 0.421  | 0.114   | 0.097   |
> >
> > Based on these results, we found that **LVLMs are highly sensitive to the density of the board** in the perceiving task for Gomoku. For example, Gemini1.5-pro achieved a perception accuracy of 0.864 on sparse boards (0-45 pieces) but dropped to 0.412 on nearly full boards (181-225 pieces). In contrast, Claude-3.5-sonnet maintained a high perception accuracy of 0.774 even on dense boards.
> >
> > Interestingly, performance often declined on medium-density boards (91–135 pieces) compared to fully occupied boards, suggesting **models struggle particularly with identifying "empty" spaces** in mixed configurations of occupied and empty positions.
> >
> > For **Chess**, we extended the analysis to consider the types of pieces (King, Queen, Rook, Bishop, Knight, Pawn) on the board:
> >
> > | LVLMs              | K     | KQ    | KQR   | KQRB   | KQRBN  | KQRBNP |
> > |--------------------|-------|-------|-------|--------|--------|--------|
> > | GPT-4o            | 0.481 | 0.474 | 0.482 | 0.431  | 0.444  | 0.447  |
> > | Gemini1.5-pro     | 0.387 | 0.369 | 0.377 | 0.369  | 0.358  | 0.360  |
> > | Claude-3.5-sonnet | 0.588 | 0.512 | 0.430 | 0.376  | 0.317  | 0.296  |
> > | Qwen2-vl-7b       | 0.298 | 0.267 | 0.284 | 0.279  | 0.273  | 0.272  |
> > | Deepseek-vl-7b    | 0.014 | 0.000 | 0.000 | 0.000  | 0.000  | 0.000  |
> > | Phi3-vl           | 0.282 | 0.280 | 0.269 | 0.277  | 0.270  | 0.273  |
> > | LLaVA-1.6-7b      | 0.299 | 0.291 | 0.281 | 0.274  | 0.279  | 0.277  |
> > | InternVL2-8b      | 0.223 | 0.214 | 0.205 | 0.211  | 0.203  | 0.199  |
> >
> > Results indicate that **increasing the variety of piece types generally decreases perception accuracy**. For example, Claude-3.5-sonnet achieved 0.588 accuracy with only the King but dropped to 0.296 with all six types of pieces. The impact was less pronounced for models like GPT-4o, which showed more stable performance.

---

> > > ### Author Response · Authors · 2024-11-21
> > > **Response to Reviewer 2f5P #Q.2 - (ii)**
> > >
> > > **A.2 (ii)**
> > >
> > > **(ii) Error Categorization in the QA Task:**
> > >
> > >  We categorized errors in the QA task under the revised multiple-choice format into:
> > >
> > >   - **Incorrect Selection**: The model selected the wrong answer from the options.
> > >   - **Unexpected Format**: The model produced output deviating from the candidate options.
> > >
> > > | LVLMs              | Incorrect Selection | Unexpected Format |
> > >   |--------------------|---------------------|-------------------|
> > >   | GPT-4o            | 4541                | 44                |
> > >   | Gemini1.5-pro     | 4517                | 39                |
> > >   | Claude-3.5-sonnet | 4733                | 57                |
> > >   | Qwen2-vl-7b       | 3937                | 0                 |
> > >   | Deepseek-vl-7b    | 4479                | 3                 |
> > >   | Phi3-vl           | 4377                | 1                 |
> > >   | LLaVA-1.6-7b      | 4515                | 2                 |
> > >   | InternVL2-8b      | 4217                | 0                 |
> > >
> > >   Additionally, we analyzed errors by question type in Tic-Tac-Toe:
> > >
> > >   - **Counting**: "How many 'O' s are on the board?"
> > >   - **Position**: "What is the symbol in row A, column 1?"
> > >   - **Rule**: "Did 'X' or 'O' win the game?"
> > >
> > >   | LVLMs              | Counting | Position | Rule |
> > >   |--------------------|----------|----------|------|
> > >   | GPT-4o            | 242      | 250      | 273  |
> > >   | Gemini1.5-pro     | 233      | 245      | 242  |
> > >   | Claude-3.5-sonnet | 252      | 267      | 265  |
> > >   | Qwen2-vl-7b       | 104      | 120      | 121  |
> > >   | Deepseek-vl-7b    | 227      | 232      | 236  |
> > >   | Phi3-vl           | 147      | 171      | 167  |
> > >   | LLaVA-1.6-7b      | 205      | 219      | 189  |
> > >   | InternVL2-8b      | 117      | 103      | 105  |
> > >
> > >   Errors were evenly distributed across categories, suggesting **no specific reasoning or perception task dominates the error** patterns. This highlights a general difficulty in integrating perception, reasoning, and instruction-following.
> > >
> > >   We appreciate the reviewer’s insightful suggestions and plan to extend these analyses to all six games in our benchmark for the full release.

---

> ### Author Response · Authors · 2024-11-21
> **Response to Reviewer 2f5P #Q.3-Q.6**
>
> **Q.3** *I’m curious to better understand the motivation behind the authors’ choice of using "unbeaten rate" to measure gameplay performance?*
>
>
> **A.3** Thank you for the thoughtful question. The reason we chose to use "unbeaten rate" as the metric is that for some games, such as Tic-Tac-Toe, it is theoretically impossible to achieve a perfect win rate if the opponent plays optimally. In such scenarios, using "success rate" would not accurately reflect the model's performance, as a draw could still indicate strong gameplay capability. Therefore, we adopted "unbeaten rate" (which includes both wins and draws) to better capture the model's overall ability to avoid losing.
>
> **Q.4** *Given access to games simulators which enable repeated games, this seems like a very high variance estimator of capability.*
>
>  **A.4** Thank you for raising this important point. We understand that evaluating gameplay performance can be sensitive to specific conditions, such as the initial game state or the opponent's strategies. To ensure the robustness of our evaluation, we opted to simulate real gameplay interactions between the LVLMs and their opponents, rather than relying on a fixed set of examples. This approach provides a more dynamic and comprehensive assessment of the models' performance. To minimize potential variability and ensure reliable results, we conducted one thousand gameplay simulations for each setting and calculated the average performance metrics. This large-scale evaluation helps smooth out any inconsistencies and provides a stable estimate of the model's capabilities.
>
> **Q.5** *Moreover, it is tightly anchored to the choice of opponent implementation for each game. Did the authors consider alternatives (for example, creating a pool of opponents at different thresholds and computing Elo ratings?) Did the authors consider evaluation against non-search-based opponents?*
>
>  **A.5**  Thank you for the reviewer's thoughtful feedback. In our framework, we implemented various embedded AI-opponent agents with different skill levels. For example, we used the SunFish chess engine, which allows us to configure opponents with varying Elo ratings to simulate different levels of difficulty. Similarly, for other search-based opponents, we adjusted the search algorithm's hyperparameters to control their strength. While this flexibility in opponent design exists, our findings suggest that the primary bottleneck for current LVLMs is not the level of the opponent, but rather their fundamental ability to process visual inputs and play complete games effectively. Nonetheless, we acknowledge the value of using a broader pool of opponents, including non-search-based ones, and plan to incorporate this in future work.
>
> **Q.6** *Taking Tic-Tac-Toe as an example, presumably minimax will produce optimal play, so the best that can be achieved is a sequence of draws?*
>
>  **A.6** For Tic-Tac-Toe, it is true that an optimal minimax strategy always leads to a draw if both players play perfectly. To account for this, we used the "unbeaten rate" as a performance metric, which includes both wins and ties. This ensures that even if all outcomes are ties, the LVLM's ability to avoid losses is properly captured in the evaluation.

---

> > ### Author Response · Authors · 2024-11-21
> > **Response to Reviewer 2f5P #Q.7-Q.8**
> >
> > **Q.7** *I don’t list this point as a “weakness” since it is concurrent work, but I would suggest citing relevant pre-prints from the first half of 2024 exploring similar directions:*
> >   - *GTBench (Duan et al., 2024), which also concludes that LLMs struggle against search-based strategies (e.g. MCTS) in games like Tic-Tac-Toe.*
> >   - *GameBench (Costarelli et al., 2024), which evaluates LLMs across 9 game environments and uses visual input (for models that support this).*
> >   - *“Mastering Atari, Go, Chess and Shogi by planning with a learned model” (Nature, 2020). This is perhaps the canonical example of achieving very strong performance across multiple games (including Chess, which is used in the current submission) while also being capable of processing RGB input on Atari games by leveraging both learning and search.*
> >
> > **A.7** We sincerely thank the reviewer for pointing out the relevant concurrent and previous works that we missed during our initial literature review. The two concurrent works, GTBench (Duan et al., 2024) and GameBench (Costarelli et al., 2024), both propose game-based environments to evaluate LLM performance in reasoning tasks. However, there are several major differences between these studies and our work:
> >
> >   - Both GTBench and GameBench operate in a text-based scenario for LLMs, where an observer converts the game state into text-based prompts for the LLMs. In contrast, our approach directly provides the raw visual input (e.g., screenshots of the game state) to LVLMs, enabling them to perceive and reason about the game state autonomously without relying on textual descriptions.
> >
> >   - GTBench primarily explores the effects of different prompt engineering techniques (e.g., Chain-of-Thought and Tree-of-Thought prompting) on LLM reasoning in game scenarios. On the other hand, our work focuses on evaluating the misalignment between the visual and language reasoning capabilities of various LVLMs, which has not been the central focus of either GTBench or GameBench.
> >
> >   - GTBench employs an LLM-vs-LLM competition to measure a Normalized Relative Advantage across different models. In contrast, our work investigates the gap between LVLMs and traditional search-based AI strategies, aiming to identify whether current LVLMs can achieve comparable performance to such methods.
> >
> > **We have incorporated these discussions into the updated version of the manuscript in the related work section to better contextualize our contributions.**
> >
> >
> > **Q.8** *Minor typos:*
> >   - *L379 (missing parenthesis).*
> >   - *Claude “soonet” is presumably intended to be "sonnet" in Table 4.1 (this typo appears in multiple places in the paper).*
> >
> > **A.8**  Thank you for your careful review and for pointing out these typos. We have carefully proofread the manuscript and revised all identified errors, including the missing parenthesis and the correction of "soonet" to "sonnet" and other instances throughout the paper. The updated version has been uploaded, with all corrections highlighted in blue.

---

> ### Author Response · Authors · 2024-11-23
> **Follow up on Discussion**
>
> Dear Reviewer 2f5P ,
> Thank you for your valuable feedback on our submission. We hope that our responses have addressed your concerns. Please let us know if there are any remaining issues or further questions, and we will be happy to provide additional clarifications. We appreciate your time and effort during the discussion period.

---

> > ### Comment · Reviewer_2f5P · 2024-11-24
> >
> > I thank the authors for their extensive efforts to address my questions and concerns.
> >
> > The non-monotonic perception performance vs board complexity results are interesting - I would not have predicted this result in advance.
> >
> > On the basis of my review and the evidence presented in the rebuttal, I plan to argue for acceptance.

---

> > > ### Author Response · Authors · 2024-11-25
> > >
> > > We greatly appreciate the reviewer's positive feedback on our work and response, and we thank you again for the constructive comments during the review process, which have greatly helped us improve our work.

---

### Official Review · Reviewer_ciTF · 2024-11-03

**Soundness:** 3
**Presentation:** 3
**Contribution:** 3
**Rating:** 6
**Confidence:** 4

**Summary:**

To address the lack of a comprehensive evaluation framework for assessing Large Vision-Language Models (LVLMs), this study proposes a framework called LVLM-Playground with game-base evaluation. This study collects six games: Tic-Tac-Toe, Reversi, Minesweeper, Gomoku, Sudoku, and Chess, to evaluate both powerful open-source and closed-source models across four tasks: Perceiving, Question Answering, Rule Following, and End-to-End (E2E) Playing. The framework provides insights into the performance shortcomings and iterative improvements of LVLMs. The study also finds that a small amount of data from games can enhance the model's reasoning abilities significantly.

**Strengths:**

1. This study collects multiple games to evaluate the multimodal large models' abilities in perception, reasoning, decision-making, and adversary, and it also visualizes the games.

2. This study evaluates the top-performing models and presents multiple findings, providing guidance for future improvements of multimodal models.

3. The authors proposed a set of metrics to evaluate the difficulty of test games in terms of Perception, Reasoning, Decision, and Adversary, which can aid in assessing game-based evaluations when selecting games in the future.

**Weaknesses:**

1. I noticed that in **Findings 6. Gameplay Data Could Enhance LVLMs' Reasoning During SFT**," the early work MiniGPT-4 was used as a base model. Since the SFT stage of MiniGPT-4 only used detail description data, it is difficult to prove that the performance gains are due to the VQA data from the game, rather than other factors like data format. Therefore, I believe that the experiments in Table 5 are not sufficient to fully support your conclusions. I suggest supplementing this with more robust base models, such as LLaVA-1.5 or LLaVA-one-vision.

2. In my opinion, these types of games are all based on dense grid patterns, and the format used to output game states might be too challenging for current multimodal large models. This could potentially limit the evaluation of the model's decision-making abilities.

**Questions:**

1. I have noticed that various models are currently unable to succeed under the E2E play setting. However, E2E play is a very important setting in LVLM-Playground. Would it be possible to introduce intermediate evaluation metrics to assess model performance?

---

> ### Author Response · Authors · 2024-11-21
> **Response to Reviewer ciTF #Q.1**
>
> **Q1.** *I noticed that in Findings 6. "Gameplay Data Could Enhance LVLMs' Reasoning During SFT," the early work MiniGPT-4 was used as a base model. Since the SFT stage of MiniGPT-4 only used detailed description data, it is difficult to prove that the performance gains are due to the VQA data from the game, rather than other factors like data format. Therefore, I believe that the experiments in Table 5 are not sufficient to fully support your conclusions. I suggest supplementing this with more robust base models, such as LLaVA-1.5 or LLaVA-OneVision.*
>
> **A.1** We thank the reviewer for their valuable feedback. We selected MiniGPT-4 as the base model for its simplicity and ease of implementation during our preliminary exploration. However, after considering the reviewer's insightful comments, we recognized the limitations of this choice and conducted additional experiments with a more robust base model, LLaVA-OneVision, to further investigate the impact of game-based instruction data for SFT on LVLMs.
>
> LLaVA-OneVision employs three training stages: Language-Image Alignment, High-Quality Knowledge Learning, and Visual Instruction Tuning. However, since the intermediate models from these stages are not publicly available and the whole instruction tuning process involves nearly 5M instruction data, replicating such large-scale instruction tuning experiments within the response period was not feasible. Instead, we conducted a preliminary exploration by fine-tuning the final LLaVA-OneVision model using game-based instruction data. The detailed configurations of our experiment are as follows:
>
> - **Model:** LLaVA-OneVision-0.5B
> - **Adaptor:** `LoRA(r=32, lora_alpha=32, lora_dropout=0.1, target_modules=['q_pro', 'k_proj', 'v_proj', 'o_proj'])`
> - **Learning Rate:** `2×10^-6 `
> - **Trainable Module:** LoRA adaptor; all other modules such as Vision-encoder, Vision-Language Projectors are fixed.
> - **Instruction Data:** 2k × 6 Games; 5k × 6 Games; 10k × 6 Games.
> - **Epoch:** 1
>
> We then evaluated the fine-tuned model on the MME benchmark, which encompasses a variety of multi-disciplinary tasks for evaluating LVLMs. The performance differences compared to the original LLaVA-OneVision (baseline) model are summarized below:
>
> | LVLM             | Base     | 2k × 6 Games | 5k × 6 Games | 10k × 6 Games |
> |-------------------|----------|--------------|--------------|---------------|
> | LLaVA-OneVision  | 240/1238 | 241/1238     | 238/1238     | 233/1238      |
>
> From the experiment results, we did not observe improvements in the MME benchmark. This could be due to several reasons:
>
> - **Extensive Pre-training:** LLaVA-OneVision was extensively pre-trained across three stages with nearly 10M high-quality vision data, making it challenging to observe notable performance gains from a relatively small-scale fine-tuning with game-based instructions.
> - **Intermediate Models Unavailable:** Since the intermediate models from the training stages were unavailable, we directly performed SFT using the game-based instructions on the final fully fine-tuned LLaVA-OneVision model, which might not have been optimal and could have even degraded performance.
> - **Training Scope and Resources:** LLaVA-OneVision employs a very large batch size of 512 during its training, where the full model, including the projectors, vision encoders, and other components, is trainable. In contrast, due to resource constraints, our fine-tuning experiments were limited to updating only lightweight LoRA parameters while keeping the rest of the model frozen. This difference in training might have further limited the performance impact.
>
> Given these observations and the reviewer's valid concerns, we acknowledge that Finding 6 in the original paper requires more in-depth analysis to draw definitive conclusions. To avoid potential confusion, we have removed this finding in the updated version of the paper. We leave a more detailed and comprehensive exploration of this topic for future work.
>
> We sincerely thank the reviewer for pointing out this important issue, which has helped us refine our analysis and future directions. Importantly, the removal of Finding 6 does not affect the overall contribution of LVLM-Playground, as the benchmark's core strengths lie in the comprehensive evaluation framework for perception, reasoning, decision-making, and adversary abilities across structured game environments. We believe these contributions remain robust and we hope the benchmark could be used for advancing the evaluation and development of LVLMs.

---

> > ### Author Response · Authors · 2024-11-21
> > **Response to Reviewer ciTF #Q.2**
> >
> > **Q.2** *In my opinion, these types of games are all based on dense grid patterns, and the format used to output game states might be too challenging for current multimodal large models. This could potentially limit the evaluation of the model's decision-making abilities.*
> >
> > **A.2** Thank you for raising this insightful concern. We agree that the dense grid patterns present in these types of games can pose a significant challenge for current large multimodal models, particularly in processing visual inputs and translating them into effective decision-making. However, this challenge is one of the key motivations behind our work. While LVLMs have demonstrated impressive performance on natural scenes that are well-represented in their training data, less representative scenarios, such as grid-like structures in these games, remain underexplored. Evaluating models in such environments helps identify their limitations and highlights areas for further improvement.
> >
> >   To ensure a balanced and comprehensive evaluation, our benchmark includes **intermediate tasks** (such as perceiving, question answering, and rule-following) that decompose the decision-making process into smaller, manageable components. These tasks allow us to isolate and evaluate specific capabilities of LVLMs, such as accurately perceiving the game state, reasoning about rules, and executing appropriate actions, before engaging in end-to-end gameplay.
> >
> >   Furthermore, to mitigate the inherent challenges posed by dense grid structures, we provide **clear instructions and standardized input-output formats** for each task. For example, in the revised version we have uploaded, we use a multiple-choice format for question-answering tasks, ensuring that models focus on reasoning and perception **without being penalized for formatting issues** or verbose outputs.
> >
> >   While we acknowledge that the dense grid-based structure could potentially limit the evaluation of higher-level decision-making abilities for some models, we believe that addressing these challenges is essential for advancing the field. We hope that our benchmark can help reveal the current limitations of LVLMs in such structured settings and serve as a foundation for fostering further advancements in both model development and evaluation methodologies.

---

> > > ### Author Response · Authors · 2024-11-21
> > > **Response to Reviewer ciTF #Q.3**
> > >
> > > **Q.3** *I have noticed that various models are currently unable to succeed under the E2E play setting. However, E2E play is a very important setting in LVLM-Playground. Would it be possible to introduce intermediate evaluation metrics to assess model performance?*
> > >
> > > **A.3**  We thank the reviewer for the thoughtful comment. We agree that the E2E gameplay setting is a critical aspect of LVLM-Playground, as it aims to evaluate the overall capabilities of LVLMs, including perception, reasoning, decision-making, and other skills. Given the poor performance of existing models in the E2E gameplay setting, we developed fundamental tasks such as perception, question-answering, and rule-following to evaluate these abilities from multiple aspects. The results of these tasks have provided some insights into the specific strengths and weaknesses of the models.
> > >
> > >   However, we also agree with the reviewer that introducing an intermediate metric to evaluate and compare different models in the E2E setting is important. Even though current models struggle to complete full gameplay, there are still measurable performance differences in their partial success. To address this, we introduced an intermediate evaluation metric that measures the **average number of successful steps** taken by a model during gameplay.
> > >
> > >   In the E2E gameplay setting, each model has a maximum of three attempts to make a valid move per round. If the model fails to follow the rules or makes a valid move after three trials, the game is terminated and marked as a failure. Under this metric, models that can consistently follow rules and make valid moves will achieve longer sequences of successful steps. We conducted 1,000 matches for each model and measured the average number of successful steps as follows:
> > >
> > >   | LVLMs                | TicTacToe | Reversi | Sudoku | Minesweeper | Gomoku | Chess |
> > >   |----------------------|-----------|---------|--------|-------------|--------|-------|
> > >   | GPT-4o              | 2.35      | 1.53    | 2.17   | 2.84        | 5.63   | 2.37  |
> > >   | Gemini1.5-pro       | 2.14      | 1.42    | 2.53   | 2.77        | 5.89   | 2.19  |
> > >   | Claude-3.5-sonnet   | 2.33      | 1.37    | 2.34   | 2.91        | 6.02   | 2.58  |
> > >   | Qwen2-vl-7b         | 1.14      | 0.83    | 0.79   | 1.78        | 2.38   | 0.87  |
> > >   | Deepseek-vl-7b      | 1.00      | 0.04    | 0.09   | 1.14        | 1.17   | 0.09  |
> > >   | Phi3-vl             | 1.52      | 0.07    | 0.21   | 1.29        | 1.46   | 0.12  |
> > >   | LLaVA-1.6-7b        | 1.47      | 0.13    | 0.17   | 1.36        | 1.54   | 0.19  |
> > >   | InternVL2-8b        | 1.00      | 0.26    | 0.43   | 1.51        | 2.10   | 0.23  |
> > >   | Random              | 2.36      | 1.15    | 1.22   | 3.45        | 27.01  | -     |
> > >
> > >   Specifically, taking the Tic-Tac-Toe game as an example, we implemented a random baseline by setting a random player A versus a search-based opponent B. At each turn, A randomly selects a position on the grid without checking whether it is empty or occupied, while B, using a search-based method, always follows the rules. If A violates the rules, the game is terminated. We calculated the average success steps of A, i.e., the average number of A's stones successfully placed on the board. After 1,000 matches, we found the random baseline to be 2.36. We then tested all LVLM-based methods under this setting and found that no model exceeded this random baseline on the Tic-Tac-Toe game, indicating that the existing LVLMs struggle to play the game based on the visual input of the game board.
> > >
> > >  However, this metric still revealed some meaningful trends: commercial models demonstrated better performance than open-source ones, and certain models, such as DeepSeek-VL and InternVL, could only place the first stone but failed to follow the rules in subsequent rounds. We then extended this intermediate evaluation metric to all other games in the benchmark. The results, as shown in the above table, highlight similar trends across different games. These findings have been incorporated into the updated version of the manuscript. We believe this intermediate metric provides some insights into the partial successes and limitations of current LVLMs in E2E gameplay.

---

> ### Author Response · Authors · 2024-11-23
> **Follow up on Discussion**
>
> Dear Reviewer ciTF,
> Thank you for your valuable feedback on our submission. We hope that our responses have addressed your concerns. Please let us know if there are any remaining issues or further questions, and we will be happy to provide additional clarifications. We appreciate your time and effort during the discussion period.

---

> > ### Comment · Reviewer_ciTF · 2024-11-24
> > **The response to the authors**
> >
> > Thank you for your detailed response to my question. Regarding the new design of intermediate metrics, I think it is a good approach, as it essentially means that each model performs "best of N" at each step.
> >
> > In addition, I would suggest considering a metric that evaluates whether the model's actions are legal. This could help assess whether the model has properly understood the information or followed the instructions.
> >
> > Moreover, for simpler games, it might be useful to design some basic baseline models and have them compete with your model to further evaluate its performance.

---

> > > ### Author Response · Authors · 2024-11-25
> > >
> > > We sincerely thank the reviewer for the positive feedback on our work and our response.
> > >
> > > Regarding the new questions, we have addressed them as follows:
> > >
> > > **NQ.1** *I would suggest considering a metric that evaluates whether the model's actions are legal. This could help assess whether the model has properly understood the information or followed the instructions.*
> > >
> > > **A.1** To address the reviewer's suggestion about introducing a metric to evaluate whether the model's actions are legal, we conducted additional experiments using the Tic-Tac-Toe game as an example. Specifically, we analyzed the errors made by LVLMs during End-to-End gameplay to assess their ability to understand the game rules and follow the instructions. We categorized the errors into the following two types:
> > >
> > > - **Format Errors:** This category includes cases where the LVLM fails to generate the next move in a parsable format, such as specifying the cell as A1, A3, or B2 (order and case insensitive). If the LVLM's response is unparsable, it will be marked as a format error, which reflects limitations in the model's instruction-following capability.
> > > - **Reasoning Errors:** This includes moves that are correctly formatted and parsable by the game simulator but are invalid within the context of the game. For instance, in Tic-Tac-Toe, any move outside the 9 valid cells (A1, A2, A3, B1, B2, B3, C1, C2, C3) is classified as an invalid movement. Additionally, errors such as repeatedly occupying an already filled cell or selecting an out-of-board position are included in this category. These errors reflect the reasoning ability and rule understanding limitations of the LVLMs.
> > >
> > > We analyzed 1,000 rounds of gameplay and recorded the frequency of errors that caused the termination of the game. The results are presented below:
> > >
> > > | LVLM               | Format Errors | Reasoning Errors |
> > > |---------------------|---------------|-------------------|
> > > | GPT-4o             | 43 | 957 |
> > > | Gemini1.5-pro      | 36 | 964 |
> > > | Claude-3.5-sonnet  | 62 | 938 |
> > > | Qwen2-vl-7b        | 19 |981|
> > > | Deepseek-vl-7b     |39|961|
> > > | Phi3-vl            |29|971|
> > > | LLaVA-1.6-7b       |17|983|
> > > | InternVL2-8b       |14|986|
> > >
> > > From the results, we observed that commercial models tend to exhibit more format errors (due to refusal to follow the instructions), aligning with the findings in the main paper. However, most errors were dominated by reasoning-related issues, highlighting the limitations of existing LVLMs in game-based scenarios.
> > >
> > > We appreciate the reviewer's valuable suggestions, and we will extend this analysis to all six games included in the LVLM-Playground, incorporating more fine-grained error categorizations based on the specific game rules in our final release. We will also provide the evaluation logs to facilitate further research and analysis.
> > >
> > > **NQ.2** *For simpler games, it might be useful to design some basic baseline models and have them compete with your model to further evaluate its performance.*
> > >
> > > **A.2** We appreciate the reviewer's insightful suggestion. For simpler games, such as Tic-Tac-Toe, we currently use search-based AI opponents like MiniMax, which produce optimal strategies. As a result, the best outcome LVLMs can achieve against such opponents is typically a draw. We agree with the reviewer that introducing more basic baseline models could provide additional insghts into the performance of LVLMs. We plan to explore this idea further and incorporate such baseline models when publishing the project.
> > >
> > >
> > >
> > > We would like to sincerely thank the reviewer again for the constructive suggestions during the review process and for the kind support in raising the rating score. We will further refine and polish our paper based on the reviewer's comments when formally publishing our work.

---

### Author Response · Authors · 2024-11-21
**General Response to All Reviewers**

We would like to express our sincere gratitude to all the reviewers and ACs for their time and effort in providing detailed feedback on our paper. We deeply appreciate the thoughtful and constructive comments, as well as the recognition of the strength of our work. We are particularly encouraged by the reviewers' acknowledgment of the novelty, significance, and clarity of our contributions. For example:

- *This study evaluates the top-performing models and presents multiple findings, providing **guidance for future improvements** of multimodal models.* (Reviewer ciTF)
- *By introducing a visual component into simple game-based evaluation, the authors open up **an interesting avenue** of possibilities for **efficient evaluation** of model capabilities.* (Reviewer 2f5P)
- *The idea of using structured game environments to evaluate vision-language models is somewhat **novel**.* (Reviewer giEs)
- *LVLM-Playground provides a **novel**, game-based approach to evaluating LVLMs, offering a more **dynamic and comprehensive assessment** than traditional benchmarks.* (Reviewer iEip)

We also sincerely thank the reviewers for their constructive and thoughtful feedback on the weaknesses and potential improvements of this work. We have carefully addressed all the questions and concerns raised in the reviews. Additionally, we have uploaded a revised version of the manuscript, where all modifications are highlighted in **BLUE**. A summary of these modifications is as follows:

- **Expanded Related Work.** We have expanded the related work section to include some discussions on the major differences and enhancements of our approach compared to some concurrent and prior works we missed in our initial version. **(reviewer 2f5P)**

- **Additional Human Study.** We have conducted a further human study to obtain ratings of the required abilities for each game across four aspects: perception, reasoning, decision-making, and adversary skills. These ratings provide additional validation of the difficulty levels defined in our benchmark. **(reviewer giEs)**

- **More Quantitative Analysis.** We have performed an additional quantitative analysis to examine how board density and piece types impact model perception and instruction-following capabilities. **(reviewer 2f5P)**

- **Revised Question-Answering Task.** We have revised the question-answering task, converting it from an open-ended format to a multiple-choice format. This modification eliminates penalties associated with verbose or unexpected output formats, decouples instruction-following abilities from perception and reasoning, and better reflects the model's core competencies. **(reviewers giEs, iEip)**

- **Error Analysis on QA Task.** Based on the revised multiple-choice format for the question-answering task, we have conducted additional experiments to explore error distributions by question type and response errors. This analysis provides deeper insights into the challenges faced by models and highlights specific areas for improvement. **(reviewer 2f5P)**

- **Game-based Data for SFT.** We have conducted a more detailed exploration of using game-based instruction data for supervised fine-tuning of LVLMs and evaluated its impact on generic VQA performance. Then refined our findings sections. **(reviewers ciTF, giEs)**

- **Intermediate Evaluation Metric for E2E Games.** Considering that current LVLMs are not yet able to complete games end-to-end, we introduced an intermediate evaluation metric that measures the length of valid steps taken by the LVLMs during gameplay. This metric provides a meaningful reflection on their overall gameplay capability and highlights their current limitations in E2E scenarios. **(reviewer ciTF)**

- **Fixing Typos.** We have carefully proofread the manuscript and fixed typographical errors, formatting inconsistencies, and other minor issues to improve the overall clarity and presentation of the paper.

We hope these clarifications address the reviewers' concerns. **We are open to any further comments or questions and look forward to hearing from the reviewers during the following discussion period.** Thank you once again for your valuable feedback.

---

### Meta-Review · Area_Chair_k16f · 2024-12-19

**Metareview:**

This paper introduces LVLM-Playground, a game-based benchmark designed to evaluate the capabilities of Large Vision-Language Models (LVLMs). Featuring six diverse games and four task categories—Perceiving, Question Answering, Rule Following, and End-to-End Playing—this benchmark extends evaluation beyond the constraints of traditional Visual Question Answering (VQA) benchmarks. Experimental results highlight key limitations of LVLMs, including challenges in generating long outputs, detailed perception, and adherence to instructions, potentially attributed to Reinforcement Learning with Human Feedback (RLHF). Additionally, the findings demonstrate that incorporating gameplay data during fine-tuning can enhance the models' reasoning abilities. A great amount of discussion regarding the usefulness of the new benchmark among reviewers, and the AC share the common enthusiasm with them. I recommend accept.

**Additional Comments On Reviewer Discussion:**

A great amount of insightful discussions; thank you, everyone!

---

### Decision · Program_Chairs · 2025-01-22

Accept (Poster)